# Regulation of striatal cells and goal-directed behavior by cerebellar outputs

Le Xiao[1], Caroline Bornmann[1], Laetitia Hatstatt-Burklé[1] & Peter Scheiffele [1]

The cerebellum integrates descending motor commands and sensory information to generate predictions and detect errors during ongoing behaviors. Cerebellar computation has been proposed to control motor but also non-motor behaviors, including reward expectation and cognitive flexibility. However, the organization and functional contribution of cerebellar output channels are incompletely understood. Here, we elaborate the cell-type specificity of a broad connectivity matrix from the deep cerebellar nuclei (DCN) to the dorsal striatum in mice. Cerebello-striatal connections arise from all deep cerebellar subnuclei and are relayed through intralaminar thalamic nuclei (ILN). In the dorsal striatum, these connections target medium spiny neurons, but also ChAT-positive interneurons, a class of tonically active interneurons implicated in shifting and updating behavioral strategies. Chemogenetic silencing of cerebello-striatal connectivity modifies function of striatal ChAT-positive interneurons. We propose that cerebello-striatal connections relay cerebellar computation to striatal circuits for goal-directed behaviors.

---

[1] Biozentrum, University of Basel, 4056 Basel, Switzerland. Correspondence and requests for materials should be addressed to P.S. (email: peter.scheiffele@unibas.ch)

The cerebellum is a major site of multisensory integration that controls sensorimotor function[1,2]. Current models for cerebellar function pose that cerebellar computation integrates motor commands from the cerebrum, proprioceptive information and other sensory inputs from the periphery. Any mismatch between the predicted and actual sensory information triggers an error signal that is forwarded to update and adjust ongoing behaviors[3]. Cerebellar computation has been most extensively studied in the context of sensorimotor functions but more recent work highlighted critical contributions of the cerebellum to non-motor functions in rodents, non-human primates, and humans[4–8]. These include cognitive and affective processes that are altered in neurodevelopmental disorders, adding to the growing body of evidence that cerebellar defects contribute to the etiology of such diseases[9–12].

The cerebellum consists of densely packed parallel microcircuits organized across the lobules of the cerebellar cortex[13–15]. Purkinje cell ensembles organized in cerebellar lobules and rostro-caudal stripes process information related to different sensory modalities and body parts[13]. Purkinje cell axons converge onto the deep cerebellar nuclei (DCN), and elicit a time-locked firing response when Purkinje cells are synchronized[16]. DCN-derived axons represent the only cerebellar outputs, and their function has mostly been considered to be directed to the brain stem and to cortical regions via thalamic relay stations. For example, cerebellum-derived information about the internal state and/or behavioral errors is routed to primary motor cortex to adjust movement control[17]. However, the diversity and anatomical organization of thalamic relay cells and targets of sub-cortical cerebellar outputs remain poorly understood.

More recent studies provided evidence for disynaptic cerebello-striatal connection emerging from the DCN[18]. Connectivity between the lateral (dentate) deep cerebellar nucleus and basal ganglia via intralaminar nuclei (ILN) of the thalamus was first reported using chemical tracers in rats[19]. Anatomical tracing with multisynaptically transported viruses in primates provided further support for communication between cerebellum and basal ganglia[20] and functional studies in mice provided physiological evidence for striatal short-latency responses upon exogenous stimulation of neurons in the lateral DCN[21]. Moreover, it was demonstrated that silencing ILN neurons in a mouse model with aberrantly elevated cerebellar activities can alleviate dystonia symptoms in this model[21]. These observations significantly modified the view of cerebellar output function which previously had been largely interpreted in the context of cortico-cerebellar loops. Thus, it was hypothesized that cerebello-striatal connections may serve to relay cerebellar information about a sensory state to the basal ganglia for the estimation of their value[22].

Striatal functions rely on an interplay of several cell classes. Most prominently, studied dopamine receptor 1 (D1R) and dopamine receptor 2 (D2R)-expressing medium spiny neurons (MSNs) give rise to the so-called direct and indirect pathways, which have been indicated in initiating or terminating actions, respectively. Another class of striatal cells are the tonically active cholinergic interneurons, which are marked by expression of choline acetyltransferase (ChAT interneurons)[23]. Excitatory inputs from intralaminar thalamic nuclei (ILN) regulate firing rate and intrinsic activity of striatal ChAT interneurons[24–26]. Thus, intralaminar nucleus inputs induce burst-and-pause firing patterns in striatal cholinergic cells[25,27]. The intralaminar nuclei-derived inputs to striatal ChAT interneurons are thought to carry information about sensory events which are interpreted for adjusting behavioral strategies in response to reward detection or contextual information[26,28,29]. While these properties are reminiscent of processes controlled by cerebellar computation, it is unknown whether thalamo-striatal inputs to ChAT cells are linked anatomically or functionally to cerebellar outputs. Moreover, it is unknown whether cerebello-striatal connections are unique to the outputs routed through the lateral deep cerebellar nucleus or whether they might be a general feature of all cerebellar output connectivity. Resolving these questions is central to understanding not only normal cerebellar function but also the cerebellar contribution to sensorimotor and cognitive dysfunction in disease states.

Here, we performed systematic connectivity mapping from genetically and anatomically defined neuron populations. We report that there is a dense connectivity matrix emerging from all cerebellar lobules and DCN. We find that thalamic relay neurons in the intralaminar nuclei innervate D1R and D2R-positive MSNs but also ChAT interneurons in the dorsal striatum. Finally, we use chemogenetic silencing experiments to address the functional contribution of DCN outputs to striatal ChAT neuron properties and to a striatum-dependent behavioral task.

## Results

**Mapping of cerebello-striatal connectivity via ILN.** Previous work uncovered disynaptic connections from the lateral deep cerebellar nucleus (dentate nucleus) in rodents via the thalamic centrolateral nucleus (CL) to the lateral dorsal striatum[18,19,21]. We used viral tracing in mice to extend these observations and to examine whether other DCN are disynaptically connected with the striatum. We labeled neurons in the interposed and medial DCN with AAV2-GFP (adeno-associated virus serotype 2) and examined contacts of their axons with neurons retrogradely marked from the dorsal striatum by injection of a glycoprotein-deficient rabies virus (RV-ΔG-RFP)[30] (Fig. 1a). Upon stereotaxic DCN injections of AAV2-GFP at postnatal day 18 (P18), GFP expression was detected in a large fraction of cells in the medial cerebellar nucleus and the interposed cerebellar nuclei at postnatal day 30 (P30) (Fig. 1b). No significant number of labeled cells were observed outside the primary infection side demonstrating that AAV2 variant employed is not retrogradely transported to other anatomical locations (see Supplementary Fig. 5 below for a related control experiment). In the thalamus, the vast majority of GFP-labeled axons was observed in the contralateral hemisphere. Labeled axons were most densely clustered in the intralaminar thalamic nuclei, in particular the centrolateral thalamic nucleus (CL) and in the parafascicular thalamic nucleus (PaF) (Fig. 1c, d). In both locations, varicosities in GFP-positive axons were observed in close proximity to RFP-positive somata, which had been retrogradely labeled by RV-ΔG-RFP from the dorsal striatum (Fig. 1e). By contrast, little or no GFP-labeling was detected in the posterior thalamic nuclear group (Po) and lateral habenula (LHb) (Fig. 1d).

DCN output neurons express the presynaptic excitatory marker vGluT2[31]. Consistently, vGluT2 was found concentrated at axonal contacts with the somata and distal dendrites of the CL and PaF neurons carrying the striatum-derived RFP label (Fig. 1f, g). By contrast, markers for other neurotransmitter systems (vGAT, tyrosine hydroxylase) did not overlap with GFP-positive boutons (Supplementary Fig. 1a, b). These anatomical findings identify excitatory terminals from interposed/medial DCN neurons located on somata and dendrites of intralaminar neurons that project to the dorsal striatum. In particular, the PaF was a major site where axons from the interposed DCN innervate striatum-projecting cells. Consistent with this conclusion, we find that light stimulation of DCN-derived axons expressing ChR2 results in synaptic responses in PaF neurons in acute slice preparations (Supplementary Fig. 1c).

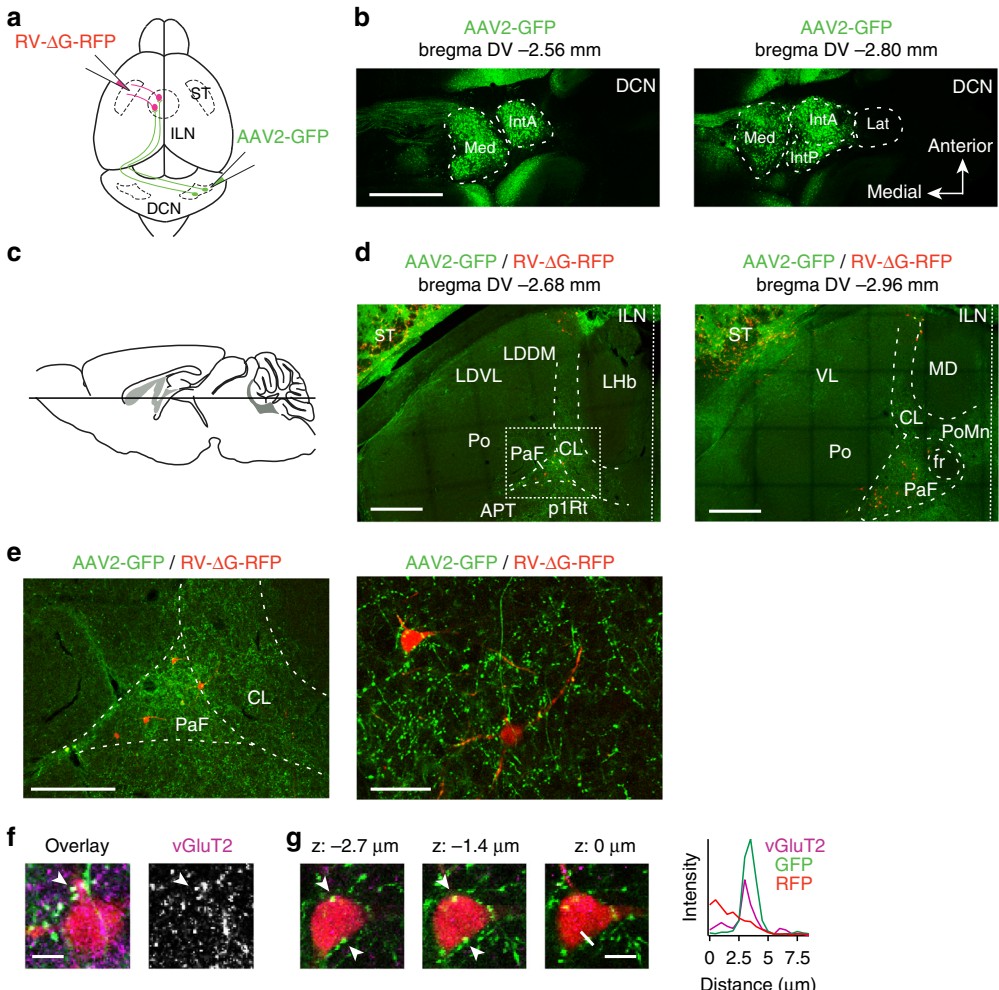

**Fig. 1** Excitatory connections from the deep cerebellar nuclei to the striatum. **a** Stereotaxic injection scheme in P18 mice. A glycoprotein-deficient retrograde rabies virus (RV-ΔG-RFP) was injected into the dorsal striatum of one hemisphere and an adeno-associated virus (serotype 2; AAV2-GFP) expressing GFP under the control of the human synapsin promoter was injected into the contralateral deep cerebellar nuclei (DCN). **b** Primary AAV2-GFP infection sites in the DCN is confined to medial cerebellar nuclei (Med) and interposed cerebellar nuclei (IntA, IntP), but does not extend to the lateral nucleus (Lat). Two dorso-ventral (DV) planes are shown with the position relative to bregma indicated. Scale bar: 1 mm. **c** Scheme of horizontal section planes used for the analysis. **d** Example images of intralaminar thalamic nuclei (ILN) at two different dorsal–ventral planes. GFP-positive axons are concentrated in several ILN substructures, including the centrolateral thalamic nucleus (CL), parafascicular thalamic nucleus (PaF) as well as the ventrolateral thalamic nucleus (VL). Note that the fluorescence seen in the striatal area (ST) is tissue autofluorescence. Thalamic neurons retrogradely labeled from the dorsal striatum with RV-ΔG-RFP are mostly observed in PaF and CL. The dashed rectangle marks the region enlarged in e. Scale bar: 500 μm. **e** Enlargement of ILN area. GFP-positive axons from DCN form varicosities onto RFP-positive neurons back-labeled from the dorsal striatum. Scale bars: 250 μm and 50 μm, respectively. **f** Some GFP-positive varicosities (arrow head) located on dendrites of RFP-positive cells are vGluT2-positive, indicating the presence of synaptic terminals. Scale bars: 10 μm. **g** Fluorescence intensity measurement demonstrating the alignment of a GFP- and vGluT2-positive varicosity apposed to a retrogradely marked RFP-expressing thalamic cell. Several consecutive optical sections from a z-stack are displayed. Scale bars: 10 μm. All experiments in this figure were replicated in three mice per condition. Anatomical annotations: APT anterior pretectal nucleus; fr fasciculus retroflexus; LDDM laterodorsal thalamic nucleus, dorsomed; LDVL laterodorsal thalamic nucleus, ventrolateral; LHb lateral habenular nucleus; MD mediodorsal thalamic nucleus; Po posterior thalamic nuclear group; PoMn posteromedian thalamic nucleus; p1Rt- prosomere 1 reticular format; ST striatum; VL ventrolateral thalamic nucleus

**DCN-striatum connections target iMSNs and ChAT interneurons**. Thalamic neurons in CL and PaF project to both the cortex and the striatum[32]. To map the axonal targets of intralaminar neurons (ILN) which receive inputs from interposed/medial DCN, we expressed in the DCN the anterograde transneuronal tracer wheat germ agglutinin (WGA) fused to cre-recombinase[33] and a cre-dependent marker (AAV2-DiO-mCherry) in the intralaminar thalamic nuclei (Fig. 2a, b). Four weeks after injection we observed abundant mCherry positive cells in CL and PaF, indicating efficient transneuronal transfer of cre-recombinase (Fig. 2c and inset). No significant signal was

observed when either one of the two tracing viruses was injected alone (Fig. 2d). Notably, ILN neurons anterogradely labeled from the cerebellum project densely to several cortical and striatal areas (Fig. 2e). In the neocortex we observed dense axon projections to motor and somatosensory cortices (M1, S1, S2; Fig. 2e). Strikingly, we also observed axonal projections to areas implicated in higher brain functions such as the cingulate cortex (A24) and the frontal associate cortex (FrA) (Fig. 2e). Axonal coverage by the DCN-ILN-derived projections in the dorsolateral and dorsomedial striatum was similar to motor and primary sensory cortices (Fig. 2f). Thus, in mice, disynaptic cerebello-striatal connections

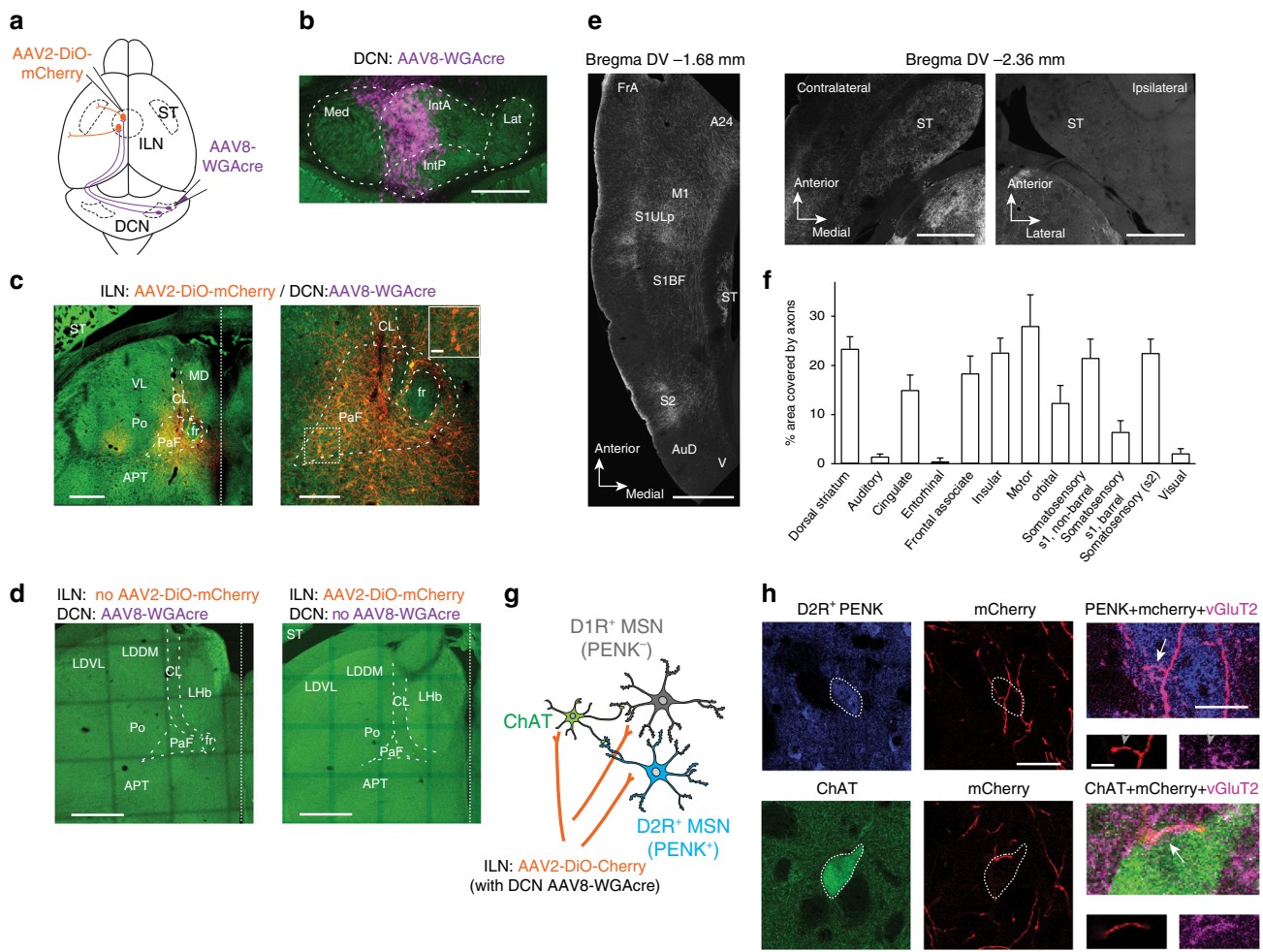

**Fig. 2** Cortical and sub-cortical targets of DCN-innervated intralaminar neurons. **a** Stereotaxic injection scheme in P18 mice. **b** The primary AAV8-WGAcre infection site is confined to medial (Med) and interposed cerebellar nuclei (IntA, IntP). Scale bar: 500 μm. **c** Overview (left) and enlargement (right) of ILN in mice four weeks post-infection. Cells expressing mCherry (orange) are observed in parafascicular nucleus (PaF) and centrolateral nucleus (CL), as well as in a few cells of the posterior thalamic nucleus (Po). Individual cell bodies are displayed in the inset derived from the area marked by a dashed square. Scale bars are 500, 200, and 20 μm as indicated. **d** Negative control section from a mouse injected with AAV8-WGAcre alone (left), and a mouse injected with AAV2-DiO-mCherry alone (right). Scale bar: 500 μm. **e** ILN neurons transneuronally labeled from the DCN project densely to several cortical and striatal areas contralateral (left and middle), but not ipsilateral (right) to the DCN injection site. Cortical areas marked are frontal association cortex (FrA), cingulate cortex, area 24a (A24), primary motor cortex (M1), upper lip (S1ULp) and barrel field (S1BF) of primary somatosensory cortex, secondary somatosensory cortex (S2), auditory cortex (AuD) and visual cortex (V). Scale bar: 1 mm. **f** Coverage by ILN-derived axons (labeled as in *a*) in cortical areas and in dorsal striatum (quantified from 14 sections derived from N = 3 mice for each area, mean ± SEM). **g** Simplified cartoon illustrating striatal medium spiny neurons (MSNs), cholinergic interneurons (ChAT) and thalamic axons (orange) emerging from ILN. **h**, Axons derived from ILN and transneuronally marked from DCN form mCherry- and vGluT2-positive contacts onto PENK-positive (D2R-positive MSNs) and ChAT-positive interneurons in the dorsal striatum. Dashed circles highlight the somata of the cells. The right panels show enlargements of the vGluT2-positive contacts. Scale bars are 30, 10, and 5 μm as indicated. All experiments in this figure were replicated in five animals. Anatomical annotations: APT anterior pretectal nucleus; fr fasciculus retroflexus; LDDM laterodorsal thalamic nucleus, dorsomed; LDVL laterodorsal thalamic nucleus, ventrolateral; LHb lateral habenular nucleus; MD mediodorsal thalamic nucleus; ST striatum; VL ventrolateral thalamic nucleus

represent a major pathway that would be predicted to strongly impact striatal target cells.

Striatal function relies on several major cell types that have unique properties, most prominently medium spiny neurons of the so-called direct (D1-receptor positive) and indirect (D2-receptor positive) pathways, as well as cholinergic (ChAT-positive) interneurons[34] (Fig. 2g). We combined immunochemical marking of striatal cell types and transneuronal labeling of ILN neurons from the interposed/medial DCN to identify targets of the cerebello-striatal connections. We found that mCherry-positive thalamic axons form glutamatergic (vGluT2-positive) terminals onto D2R-expressing MSNs (identified by immunostaining for proenkephalin[35] (PENK, Fig. 2h). Moreover, mCherry/vGluT2-positive contacts were also readily identified on ChAT-positive interneurons (Fig. 2h). Thus, the interposed/medial DCN neuron outputs through the ILN are not only relayed to multiple cortical areas but there is a dense disynaptic connection to MSNs and ChAT-positive interneurons in the striatum.

**Cerebellar input channels to striatal MSNs and ChAT neurons.**
To selectively map input channels to MSNs and ChAT-positive interneurons from DCN via the ILN, we employed multisynaptic

rabies virus tracing from genetically defined starter cells in the striatum[36,37]. Previous work has mapped monosynaptic inputs to D1R-, D2R-positive MSNs or ChAT-positive interneurons in the dorsal striatum using conditional (cre-dependent) expression of the TVA-receptor and EnvA-pseudotyped, glycoprotein (G)-deficient rabies viruses[38,39]. In separate experiments, we confirmed that the majority of monosynaptically connected cells in the thalamus are located in PaF, CL, Po, and VPM as previously reported[38,39] (see Supplementary Fig. 2a, c for monosynaptic mapping of inputs to striatal ChAT interneurons).

To reveal input channels beyond the monosynaptically connected cells we simultaneously provided viral glycoprotein (together with a GFP marker) in ILN neurons (see Fig. 3a and methods for details). Thus, viruses that are monosynaptically transported to intralaminar neurons could undergo further transsynaptic passage. Using this multisynaptic tracing approach, we revealed thalamic and DCN neurons retrogradely labeled from D1R-positive (D1R-cre line), D2R-positive (A2A-cre line) and ChAT-positive (ChAT-cre) striatal cells. Starter neurons in D1R-, A2A-cre and ChAT-cre were similarly located in the dorsolateral and dorsomedial striatum (Supplementary Fig. 3a–e). We then quantified the distribution of candidate relay neurons in the thalamus identified based on the co-expression of mCherry (derived from the rabies virus) and GFP (coexpressed with cre-recombinase that gates G expression in the ILN). mCherry/GFP double-positive cells in the thalamus were abundant in all three conditions (D1R-cre, A2A-cre mice and ChAT-cre mice; Supplementary Fig. 3f, g). PaF and CL jointly contained 76%, 70% and 55% of all relay neurons for tracing initiated from D1R-cre, A2A-cre and ChAT-cre cells, respectively (Supplementary Fig. 3f). The distribution of retrogradely labeled thalamic cells obtained with this multisynaptic configuration was very similar to what was reported for monosynaptic tracing from striatal cells (Supplementary Fig. 2a–c and refs. [38,39]). We confirmed that most GFP-positive cells in the thalamus also exhibited immunor-eactivity for a newly generated antibody to the viral G protein (Supplementary Fig. 4). Thus, most of the mCherry/GFP double-positive thalamic cells should indeed permit further transsynaptic spread of rabies viruses to ILN inputs. We observed retrograde labeling of the DCN in the contralateral but not ipsilateral hemisphere for all striatum-initiated tracings (D1R-cre: 17±5, A2A-cre: 33±8, ChAT-cre: 32±8 DCN neurons, mean ± SEM, $N = 3, 9, 7$ animals respectively).

We then assessed which deep cerebellar subnuclei gave rise to the cerebello-thalamo-striatal connections. We registered the labeled DCN neurons onto four dorsal–ventral atlas planes and assigned their DCN subnuclei associations (Fig. 3b). The back-labeled cells in the contralateral DCN exhibited dendritic morphologies of large multipolar and columnar neurons (Fig. 3c–e)[40]. Retrogradely marked cells were most abundant in the posterior interposed (IntP) and lateral (Lat) DCN (Fig. 3f). Back-labeled cells were also detected in the medial DCN, albeit with lower frequency. Notably, for the medial DCN, significantly fewer back-labeled cells were observed for the ChAT input channel as compared to A2A-cre ($p < 0.05$, chi-square test, Fig. 3f). Taken together, these experiments reveal a dense matrix of cerebellar output channels routed from all DCN via the ILN to the dorsal striatum. This connectivity targets striatal MSNs of the direct (D1R) and indirect (D2R) pathway, as well as ChAT-positive interneurons.

We next asked which cerebellar lobules provide input to the DCN-striatal connectivity. Since MSNs represent 95% of striatal neurons[23], we focused on multisynaptic retrograde tracing initiated from striatal iMSNs (A2A-cre cells) and enabled additional transsynaptic transfer of the viral tracer by expression of the viral Glycoprotein-protein in DCN neurons (Fig. 4a). Four weeks after rabies virus injection, we observed back-labeled Purkinje cells throughout all cerebellar lobules including the vermis, Crus 1 and 2 (Fig. 4b, c). The highest number of cells was observed in lobule 1/2 ($n = 21$), lobule 6 ($n = 16$) and Crus 1/Sim ($n = 13$) (Fig. 4d). Thus, most (if not all) cerebellar lobules are anatomically connected to iMSNs. It is important to note that in this multisynaptic tracing experiments (with G protein complementation in ILN as well as DCN), the back-labeled Purkinje cells are not necessarily trisynaptically connected to iMSNs. This is due to the fact that expression of G protein in ILN and DCN would also allow for synaptic transfer between cells within these structures. Nevertheless, these experiments support a broad array of cerebellar output channels to the dorsal striatum.

**Silencing of DCN impacts activity in striatal ChAT-neurons.** Striatal ChAT interneurons have been implicated in the acute modification of behavioral processes[29,41,42], a function that is reminiscent of cerebellum-dependent sensory integration and error correction. Thus, we sought to explore whether DCN projection neurons modify ChAT interneuron activity in mice. Activation of ChAT interneurons can be assessed based on the phosphorylation state of the ribosomal protein S6 on its serine 240 and 244 (pS6rp)[43,44]. We virally introduced the chemogenetic inhibitory DREADD hM4Di[45] into vGluT2-positive DCN neurons (AAV2-DiO-hM4Di x vGluT2-cre: DCN:vGluT2$^{hM4Di}$). Detailed analysis across the entire brain showed that hM4Di expression was restricted to DCN neurons, confirming that viruses were not retrogradely taken up by axons innervating the DCN (Supplementary Fig. 5). When DCN neurons were silenced by CNO injection, we detected a reduction in pS6rp immune reactivity in ChAT-immunopositive cells in the dorsomedial striatum of DCN:vGluT2$^{hM4Di}$ mice but not in CNO-treated control mice (Fig. 5a–f). Since dorsomedial ChAT neurons have been reported to be preferentially modified upon lesion of tha-lamostriatal projections[44], we compared pS6rp immune reactivity between dorsomedial and dorsolateral ChAT neurons. Notably, we did not observe a comparable reduction in pS6rp immune reactivity in dorsolateral ChAT interneurons upon chemogenetic silencing of DCN vGluT2 cells, demonstrating specific impact of DCN silencing on ChAT neurons in the dorso-medial striatum (Fig. 5g–j). This differential effect on dorsomedial versus dorso-lateral ChAT cells might be due to different basal activity states of these populations. Thus, on average, dorsolateral ChAT neurons showed lower pS6rp immune reactivity when compared to dor-somedial ChAT neurons (Fig. 5f, j, as reported previously for rats[44]).To further explore the relationship between the extent of DCN vGluT2 neuron silencing and the pS6rp marker of striatal ChAT interneuron activity, we examined the correlation of the fraction of hM4Di-mCherry-positive neurons in the IntP (that arise as a consequence of different efficiency of viral infection across experiments) and the pS6rp levels in the striatum. We observed a significant relationship for ChAT interneurons in the dorsomedial ($r^2 = 0.66$, Fig. 5f) but not the dorsolateral striatum ($r^2 < 0.0001$, Fig. 5j). Taken together, these findings demonstrate that IntP DCN outputs exert a significant activity modulation of striatal ChAT interneurons in mice.

**Cerebellar regulation of striatum-dependent behaviors.** The functional and anatomical linkage between DCN neurons and striatal ChAT interneurons suggests that inhibiting DCN neurons might modify striatum-dependent behaviors. ChAT-positive interneurons have been implicated in controlling goal-directed spatial discrimination in paradigms with changing rules[26,29,44]. We virally introduced the chemogenetic inhibitory DREADD hM4Di into vGluT2-positive neurons of the interposed and

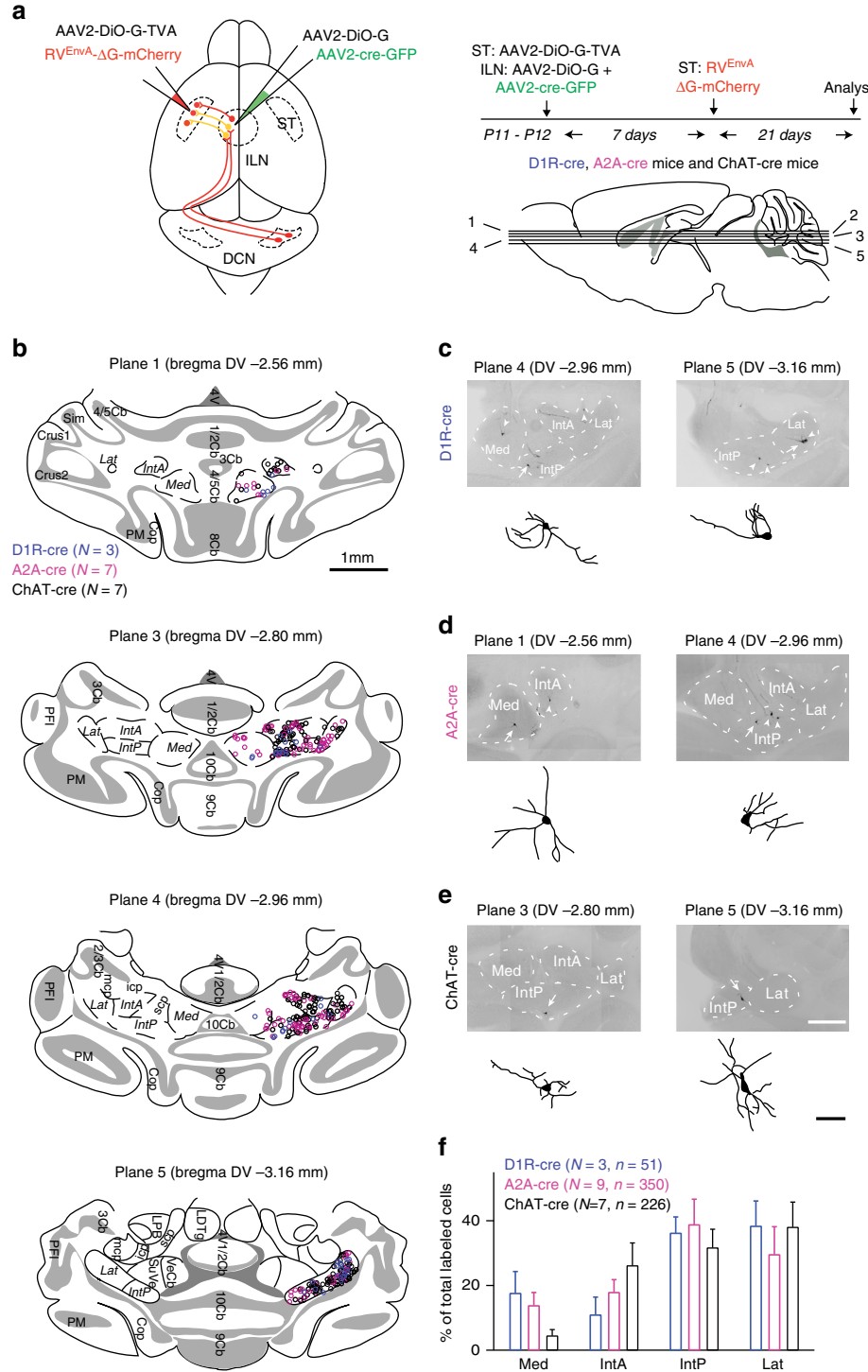

medial cerebellar nucleus at postnatal day 18 (AAV2-DiO-hM4Di x vGluT2cre: DCN:vGluT2^hM4Di, Fig. 6a, b). Location and number of hM4Di-expressing cells was confirmed by immuno-histochemistry. Viral expression reliably targeted the interposed and medial cerebellar nuclei and was only rarely detected in the lateral (dentate) deep cerebellar nucleus (Fig. 6b). Only mice with >50% IntP neurons expressing hM4Di-mCherry were included in quantifications of behavioral performance across all tasks below. CNO and its metabolite clozapine can exhibit hM4Di-

independent effects in rodents[46,47]. Thus, mice injected with AAV2-GFP, treated with CNO were used as controls for comparison.

First, we assessed the impact of CNO-mediated DCN silencing in simple motor tasks. DCN:vGluT2^hM4Di mice treated with CNO showed similar locomotion in an open field as control mice (AAV2-GFP injected into interposed/medial DCN: DCN:GFP) (Fig. 6c and Supplementary Fig. 6). Moreover, marble burying, which in mice is considered to reflect a natural repetitive behavior

**Fig. 3** Cerebello-striatal connectivity to MSNs and ChAT interneurons. **a** Stereotaxic injection and analysis scheme for multisynaptic tracing. Injections in D1R-cre (N = 3), A2A-cre mice (N = 9) and ChAT-cre mice (N = 7) are initiated at P11-P12. First, AAV2-DiO-G-TVA is injected into the dorsal striatum (ST). This results in cre-dependent expression of viral Glycoprotein (G) and the surface receptor TVA in D2R-positive or ChAT-positive striatal cells. At the same time, AAV2-DiO-G and AAV2-cre-GFP are injected into the intralaminar thalamic nuclei (ILN) to express glycoprotein in ILN neurons (see Supplementary Figs. 3 and 4 for additional information on relay cells and RV-G expression). Seven days after the first injection, EnvA-pseudotyped, glycoprotein-deficient rabies viruses are injected into the dorsal striatum. Five dorsal–ventral (DV) planes (plane 1, 2, 3, 4, 5) used for analysis illustrated. **b** Retrogradely labeled DCN neurons are registered at four dorsal–ventral planes. Cells obtained by back-labeling in D1R-cre, A2A-cre, and ChAT-cre mice are color-coded blue, magenta, and black, respectively. Data derived from tracings performed in N = 3 D1R-cre mice, N = 7 A2A-cre mice, and N = 7 ChAT-cre mice. Scale bar: 1 mm. **c**–**e** Morphology of DCN neurons retrogradely labeled in D1R-cre, A2A-cre, and ChAT-cre (arrows). *Camera lucida* drawings of example cells (arrows) reveal columnar and multipolar neuron morphologies. Scale bars: 500 μm and 100 μm, respectively. **f** Subnuclei distribution of retrogradely labeled DCN neurons. Note the overall preference for interposed nuclei (IntA, IntP) and lateral dentate nucleus in D1R-cre, A2A-cre and ChAT-cre mice. Back-labeled DCN neurons in A2A-cre mice have preferential labeling in medial dentate nucleus when compared to ChAT-cre mice ($p < 0.05$, chi-square test). N = 3 mice and n = 51 DCN cells for D1R-cre, N = 9 mice and n = 350 DCN cells for A2A-cre, and N = 7 mice and n = 226 DCN cells for ChAT-cre. Mean ± SEM

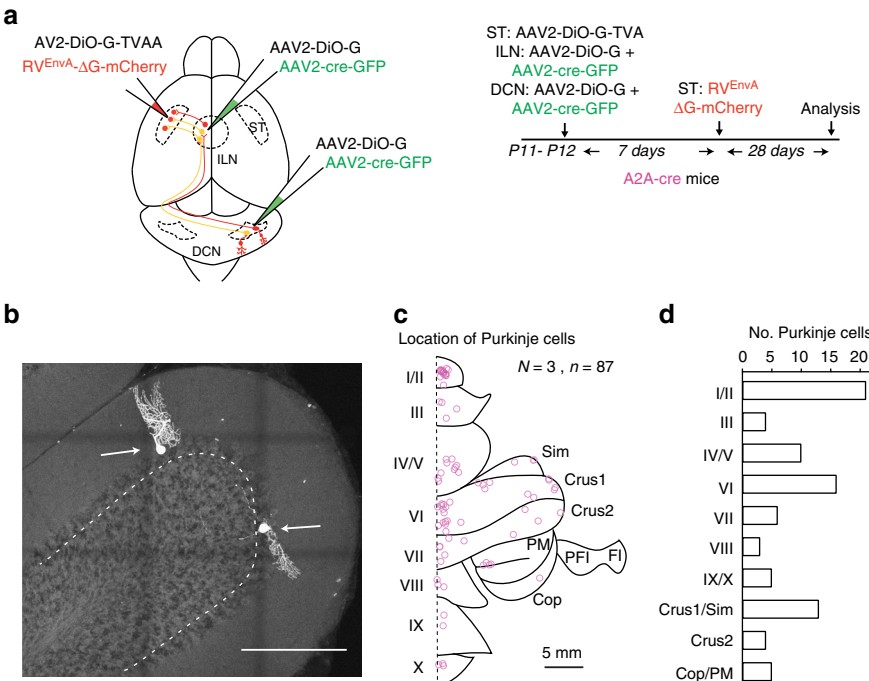

**Fig. 4** Dorsal striatal iMSNs receive information from all the cerebellar lobules. **a** Stereotaxic injection and analysis scheme. Injections in A2A-cre mice (N = 3) are initiated at P11-P12. The procedure is as in Fig. 3, except performing an additional injection of AAV2-DiO-G/AAV2-cre-GFP at P11-P12 to express viral glycoprotein in the DCN. **b** Example of retrogradely labeled Purkinje cells in Crus 1 (arrows) marked by mCherry expressed from the transsynaptic rabies virus. Scale bar: 200 μm. **c** Location of 87 retrogradely labeled Purkinje cells from tracings performed in three mice. Cell positions are mapped onto an unfolded map of the cerebellum. Scale bar: 5 mm. **d** Number of transsynaptically labeled Purkinje cells by cerebellar lobule. Note lobule 1/2, lobule 6 and Crus 1/Sim have the highest number of cells. Anatomical annotations: Cop copula of the pyramis; Crus 1 crus 1 of the ansiform lobule; Crus 2 crus 2 of the ansiform lobule; Fl flocculus; Sim simple lobule; PFl paraflocculus; PM paramedian lobule

related to digging, was unaltered (Fig. 6d). Finally, we examined spontaneous alternation behavior in a T-maze (Fig. 6e). In 10 consecutive trials, CNO-treated DCN:vGluT2hM4Di mice showed similar numbers of goal arm alternations as CNO-treated DCN: GFP control mice. These results indicate that chemogenetic silencing of a large fraction of posterior interposed and medial cerebellar nucleus neurons does not significantly modify simple motor behaviors as well as intrinsic stereotypical behaviors in mice.

We then examined a reward-seeking behavior where mice had to learn a T-maze task with changing rules. First, mice were food-deprived for 7–10 days and were then trained in the T-maze for 10 runs per day with a food reward placed in one of the goal arms[48]. In the absence of CNO during training sessions, DCN: GFP and DCN:vGluT2hM4Di mice learned to choose the baited arm at a similar rate (Fig. 6f). Once reaching the learning criterion, the importance of IntP/Med DCN neuron activity in reversal learning was assessed by CNO injection in the trained mice in two sessions on the following 2 days. Notably, CNO-injected DCN:GFP and DCN:vGluT2hM4Di mice showed very similar reversal learning on both days (Fig. 6g).

To assess the ability of mice to perform a more challenging rewarded task, we trained food-deprived mice in a forced alternation paradigm. In this test, a food reward is moved for every trial to the arm opposite to the location that was visited by the test animal in the preceding trial[49]. In the absence of CNO, DCN:GFP and DCN:vGluT2hM4Di mice learned the forced alternation task in a similar number of sessions (Fig. 6h). Once reaching the training criterion, mice were treated with CNO on the following day and tested for their performance. DCN:

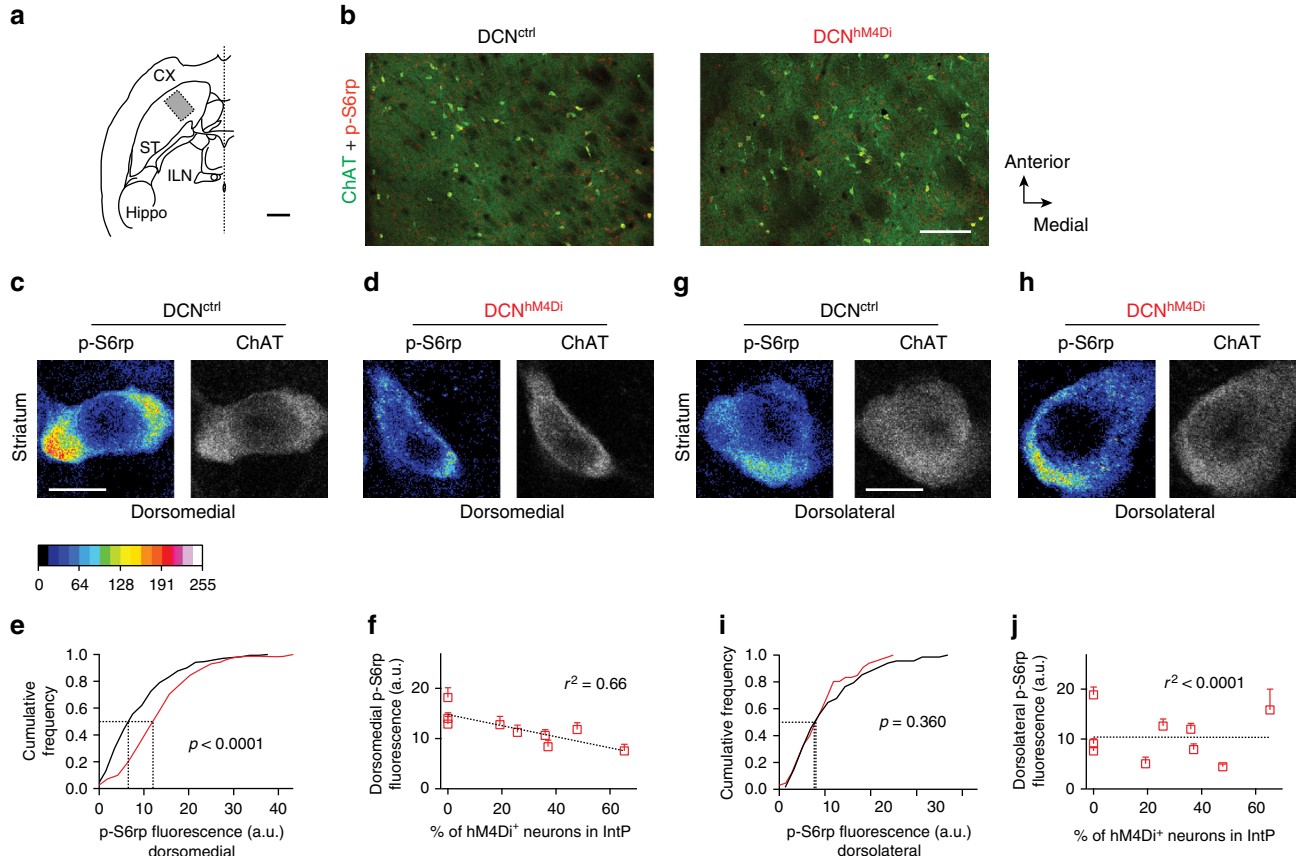

**Fig. 5** Impact of DCN silencing on ChAT interneuron activity in the striatum. **a** Schematic for overview images from dorsal striatum in horizontal sections. The square indicates the location of the image in (**b**). Anatomical annotations: CT cortex, ST striatum, ILN intralaminar thalamic nuclei, Hippo hippocampus. **b** Example overview image containing parts of dorsomedial and dorsolateral striatum from CNO-injected control mice and mice expressing hM4Di in the DCN of both hemispheres. ChAT immunoreactivity in green, p-S6rp immunoreactivity in red. Scale bar: 200 µm. **c, d** Examples of dorsomedial striatal ChAT interneurons stained with anti-ChAT and anti-p-S6rp antibodies. Medial and interposed DCN of vGluT2-cre mice were bilaterally injected at P18 with AAV2-DiO-hM4Di and mice were intra-peritoneal (i.p.) injected with CNO (2.5 mg/kg) 50 min before analysis. Immune reactivity was examined at P35 in the dorsomedial striatum $DCN^{ctrl}$ (control) and $DCN^{hM4Di}$ mouse brains. A 16-pseudocolor palette (Lookup Table) highlights the intensity of p-S6rp signal. Scale bar: 10 µm. **e** Cumulative frequency of p-S6rp fluorescence intensity in dorsomedial striatal ChAT neurons ($n = 242$ ChAT-positive neurons from $N = 3$ $DCN^{hM4Di}$ mice and from $N = 3$ control mice, $p < 0.0001$, Kolmogorov–Smirnov test). **f** p-S6rp signal intensity in dorsomedial ChAT-positive neurons in one hemisphere correlates with the percentage of hM4Di-expressing cells in the contralateral interposed deep cerebellar nucleus. Mice were bilaterally injected in the DCN. The infection rate in the DCN of each hemisphere was correlated with the p-S6rp immunereactvity in the contralateral striatum ($r^2 = 0.66$, $n = 242$ ChAT-positive neurons across six hemispheres from $N = 3$ $DCN^{hM4Di}$ mice and $N = 3$ control mice without DCN injection). Boxes display and error bars display mean ± SEM). **g, h** As in (**b, c**), but analyzing p-S6rp immune reactivity in the dorsolateral striatum. Scale bar: 10 µm. **i** Cumulative frequency of p-S6rp fluorescence intensity in dorsolateral striatal ChAT-positive neurons ($n = 134$ ChAT-positive neurons from $N = 3$ $DCN^{hM4Di}$ mice and $N = 3$ control mice, $p = 0.297$, Kolmogorov–Smirnov test). **j** p-S6rp signal intensity in dorsolateral ChAT-positive neurons show no significant correlation to the percentage of hM4Di-expressing cells in the contralateral interposed deep cerebellar nucleus ($r^2 < 0.0001$, $n = 134$ ChAT-positive neurons across six hemispheres of $N = 3$ $DCN^{hM4Di}$ mice and $N = 3$ control mice, boxes display and error bars display mean ± SEM)

$vGluT2^{hM4Di}$ mice showed a significantly worse performance ($p = 0.024$, Mann–Whitney test, two tailed) in forced alternation, while the performance of CNO-treated DCN:GFP mice was unchanged (Fig. 6i). Notably, we observed a strong correlation between the percentage of hM4Di-mCherry positive neurons in the IntP and the performance of the forced alternation task upon CNO treatment ($r^2 = 0.65$, Fig. 6j). By contrast, there was no significant correlation of behavioral performance with hM4Di-mCherry positive neurons in Med or IntA (Supplementary Fig. 7), suggesting a causal link between the activity of IntP neurons with the cognitive flexibility of mice in the forced alternation task. These findings demonstrate an important role for DCN neurons in a striatum-dependent reward-driven behavior.

DCN outputs may control striatum-dependent behavior either through cerebello-striatal connectivity or via cortico-striatal connections. To test directly the involvement of cerebello-striatal connections, we drove hM4Di expression selectively in ILN neurons that are innervated by DCN outputs (Fig. 7a, b). To this end, we stereotaxically delivered cre-dependent hM4Di (AAV2-DiO-hM4Di-mCherry) into the ILN and then selectively turned on hM4Di expression in cells contacted by DCN axons (AAV8-WGA-cre sterotaxically delivered into the DCN as described in Fig. 2). Three weeks after viral infection we observed hM4Di-mCherry expression in the ILN as well as in the ILN-derived thalamo-striatal axons (Fig. 7c, d). To selectively silence striatum-projecting axons we cannulated mice for local application of CNO in the dorsal striatum. For all mice position of the cannulae was confirmed post hoc and the spatial restriction of infusion was probed by infusion of a fluorescent dextran (see Fig. 7c for an example). Mice with striatum-specific silencing of

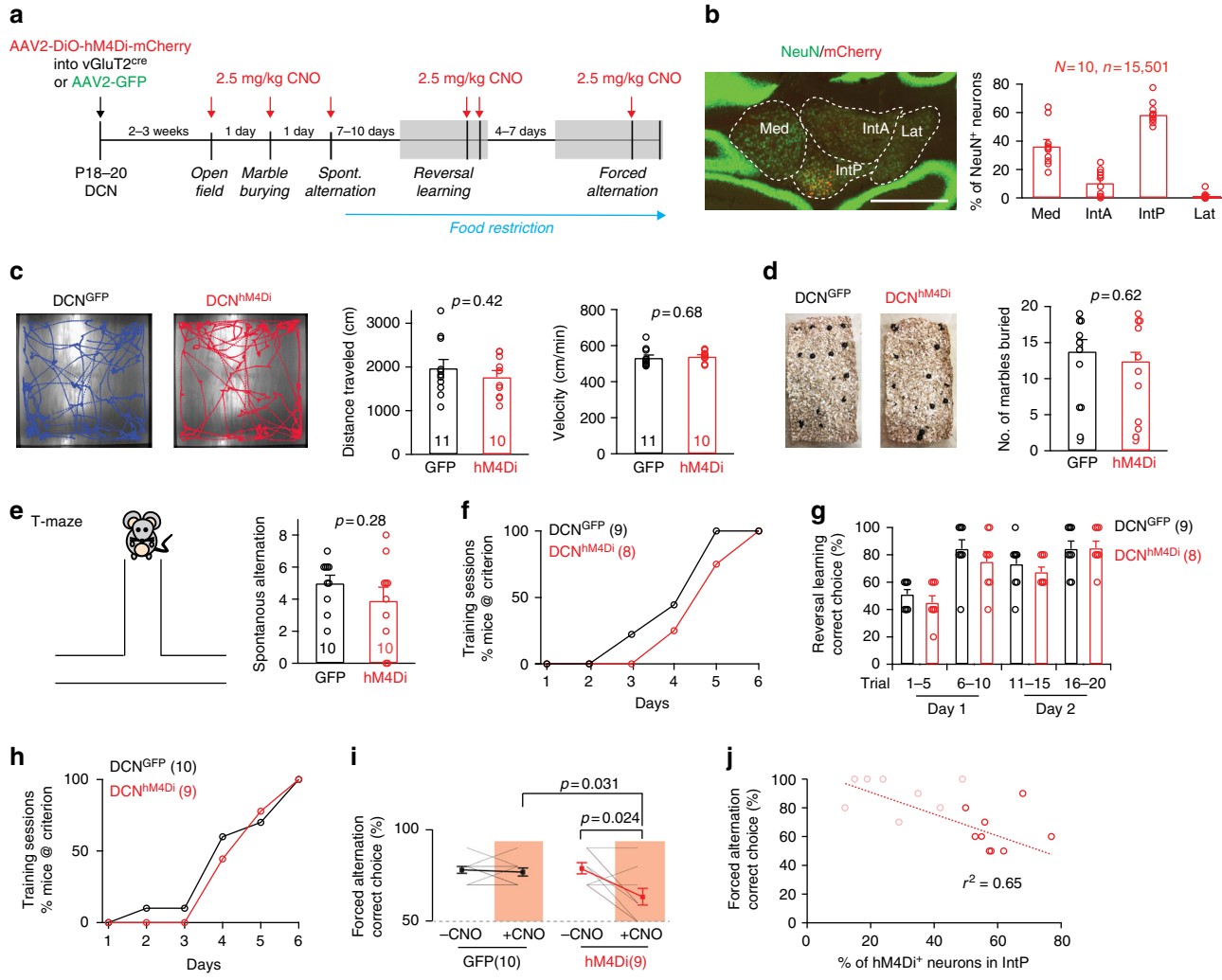

**Fig. 6** DCN neuron activity is required for goal-directed behaviors. **a** Stereotaxic injection and analysis scheme. **b** Left: Example image of AAV-DiO-hM4Di infection site in DCN of vGluT2-cre mice. Right: Quantification of hM4Di-positive neurons in each DCN subnucleus ($N = 10$ mice, mean ± SEM). Scale bar: 400 μm. **c** Open field tracks of control DCN^GFP and hM4Di-expressing DCN^hM4Di treated CNO. Right: Quantification of distance traveled in open field and velocity. ($N = 11$ and 10 mice, respectively; mean±SEM, unpaired t-test). **d** Left: example images of marble burying test. Right: number of marbles buried by DCN^GFP and DCN^hM4Di mice treated with CNO ($N = 9$ mice per group, mean ± SEM, unpaired t-test). **e** Left: Cartoon for T-maze test. Right: Number of spontaneous alternations observed in 10 consecutive trials in a T-maze without food reward. ($N = 10$ mice per group, mean ± SEM, unpaired t-test). **f** T-maze reversal learning. Mice were food-deprived and trained in a rewarded T-maze paradigm to choose one arm. Percentage of mice reaching the training criterion after each day is plotted ($N = 9$ DCN^GFP and $N = 8$ DCN^hM4Di mice). **g** T-maze reversal learning. Percentage of correct choice at Day1 and Day 2 after reversal of the rewarded side. For each day, the first 5 and second 5 trials are plotted separately to differentiate perseverative errors ($p = 0.325$, unpaired t-test) and regressive errors ($p = 0.347$, unpaired t-test; $N = 9$ DCN^GFP and $N = 8$ DCN^hM4Di mice, mean ± SEM). **h** T-maze forced alternation. The percentage of mice reaching the training criterion after each day is plotted. ($N = 10$ DCN^GFP and $N = 9$ DCN^hM4Di mice). **i** T-maze forced alternation. Performances at the last training session (-CNO) and the session after CNO treatment (+CNO) are compared. For comparison between DCN^GFP and DCN^hM4Di mice after CNO, mean ± SEM, Mann–Whitney test, two tailed. For comparison of DCN^hM4Di mice before and after CNO, mean ± SEM, Mann–Whitney test, two tailed ($N = 10$ DCN^GFP and $N = 9$ DCN^hM4Di mice). **j** Correlation of the percentage of hM4Di-positive neurons in IntP and the performance in the forced alternation task after CNO treatment in DCN^hM4Di mice. Each circle represents one animal. Thick-lined circles represent mice with >50% of IntP neurons expressing hM4Di. Thin-lined circles are mice not included (<50% of IntP neurons, $N = 8$ mice)

cerebellostriatal connectivity moved in open field with the same velocity as control mice expressing tdTomato (delivered with the same intersectional viral approach, Fig. 7e, f). However, the total distance traveled was slightly decreased due to prolonged stopping periods, indicating an alteration in the initiation of movements (Fig. 7f). When trained in the forced alternation task in the absence of CNO, hM4Di and control mice reached the training criterion in a similar number of trials (Fig. 7g, h). However, CNO-mediated striatum-specific silencing of cerebello-striatal connectivity impaired the performance in the forced alternation task (Fig. 7i). Thus, striatal silencing of axons derived

from ILN cells that receive DCN innervation replicates the behavioral phenotype in forced alternation observed with silencing of DCN neurons. In aggregate, our anatomical and functional studies reveal a critical function for cerebello-striatal connectivity in a reward-driven alternation behavior in mice.

## Discussion

In this study, we performed an in-depth mapping of cerebello-striatal connections in mice. We uncovered dense connectivity from the interposed deep cerebellar nucleus via the parafascicular

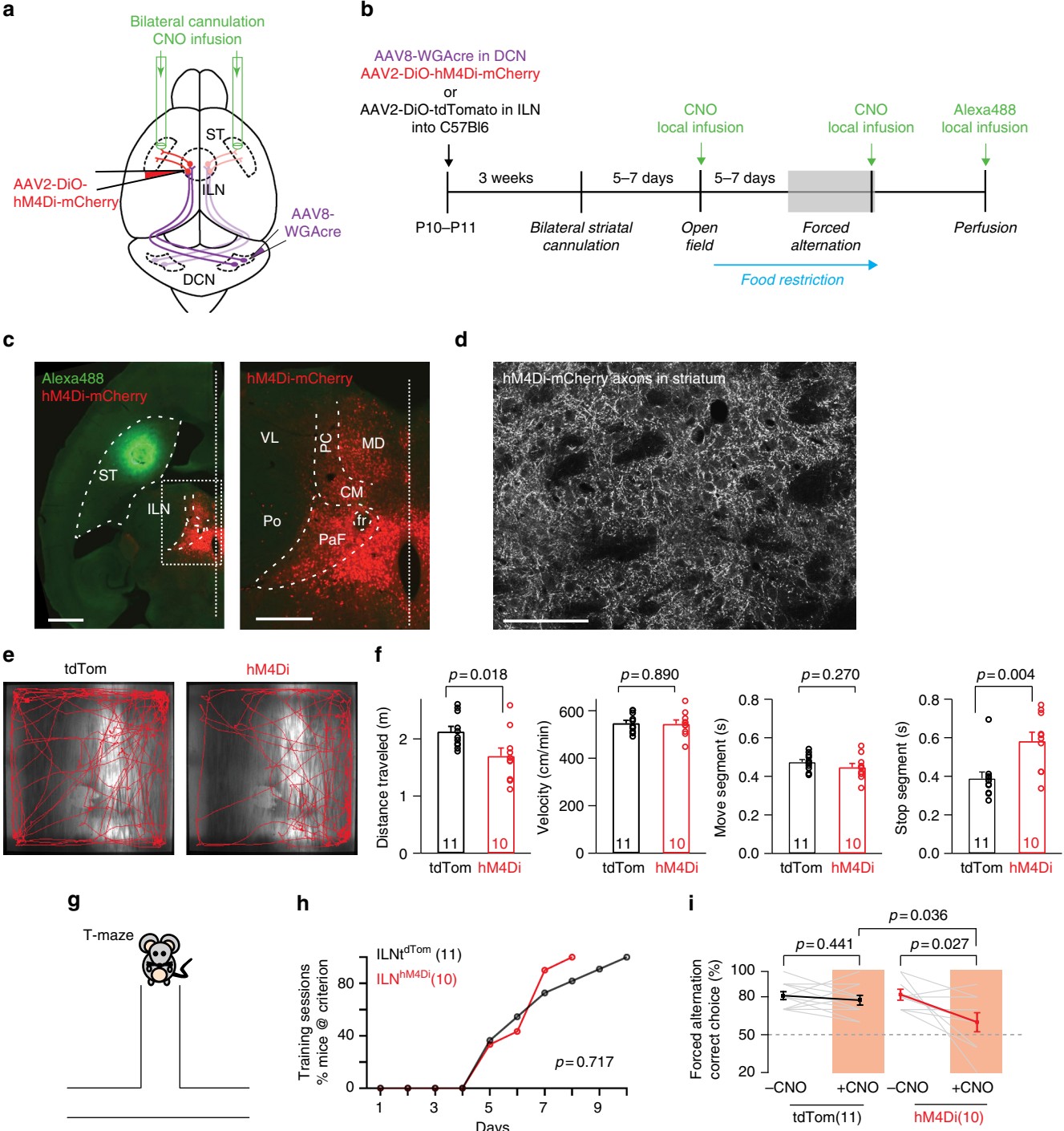

**Fig. 7** Striatum-specific silencing of cerebello-striatal connectivity. **a** Stereotaxic injection scheme for local silencing experiments. For local infusion of CNO, mice are bilaterally cannulated in the dorsal striatum. **b** Timeline and analysis scheme of local silencing experiments. See text and methods for details. **c** Overview (left) and ILN enlargement (right) of dorsal striatum and thalamic nuclei in mice at the end of the experiment. Cells expressing hM4Di-mCherry fusion protein (red) are observed in parafascicular nucleus (PaF) and centromedial nucleus (CM). The spread of Alexa488 dextran applied through the cannula in the striatum is shown in green ($N = 11$ mice). **d** High magnification image of hM4Di-mCherry positive axons in the dorsal striatum. This reveals efficient transport of the hM4Di-mCherry fusion protein into the thalamo-striatal axons. **e** Example open field tracks of control ILN$^{tdTom}$ (tdTom) and ILN$^{hM4Di}$ (hM4Di) expressing mice (targeted with intersectional viral approach described above). Mice were locally infused with CNO (30 μM, 800 nl/side) 35 min before the test. **f** Quantification of distance traveled in open field, velocity, move segments, and stop segments of control ILN$^{tdTom}$ and ILN$^{hM4Di}$ mice (intersectional viral approach described above, $N = 11$ and 10 mice, respectively; mean ± SEM, unpaired *t*-test for open field, velocity and move segments. For stop segments, Mann-Whitney test, two-tailed). **g** T-maze forced alternation. **h** Mice were trained in a forced alternation paradigm with food reward. The percentage of mice reaching the training criterion after each day is plotted. ILN$^{tdTom}$ (tdTom) and ILN$^{hM4Di}$ (hM4Di) mice (intersectional viral approach described above, $N = 11$ and 10 mice, respectively)

and centrolateral nuclei to the dorsal striatum. We find that DCN outputs are relayed to MSNs but also striatal ChAT interneurons. Finally, we demonstrate that silencing of DCN outputs modifies the function of striatal ChAT interneurons and impairs behavioral performance of mice in a striatum-dependent task. In aggregate, our findings strongly suggest that cerebello-striatal connections from interposed DCN relay cerebellar computations to several striatal cell types.

Our anatomical mapping revealed a much more extensive connectivity between DCN and the dorsolateral striatum than previously anticipated. Using anterograde and retrograde tracing we mapped connectivity emerging from the interposed cerebellar nuclei as well as a cerebello-striatal relay stations in the intralaminar nuclei of mice. Using an intersectional viral approach, we demonstrated that a wide range of neocortical and striatal areas receive disynaptic connections from the cerebellum via these intralaminar thalamic neurons. We observed dense cerebello-thalamic-cortical projections in motor and primary somatosensory cortex, two regions where previous electrophysiological studies provided evidence for functional connectivity to the cerebellum[50]. Moreover, we also discovered dense connectivity to cortical areas involved in higher cognitive functions such as the prefrontal association cortex and cingulate cortex[51]. Notably, imaging studies support functional coupling between the cerebellum and analogous cortical regions in human[12,52]. Such connections may be involved in forwarding predictions of perceptual and mental states for cognitive processes[53]. We further find that the cerebello-striatal connections are linked to most cerebellar lobules. This indicates that ILNs relay cerebellar computations related to the diverse modalities represented across the cerebellar cortex. The primate cerebellum is functionally divided into an anterior motor division (lobule I–V) and posterior non-motor division (remaining lobules)[4,6,9,13]. Human imaging studies further support differential engagement of cerebellar lobules in cognitive and motor tasks[54]. Within the striatum, we observed axons linked to cerebello-thalamic connections in both, the dorsolateral and dorsomedial striatum. Notably, along the dorsoventral and mediolateral axis, striatal sub-compartments exert distinct functions related to movement and reinforcement learning, respectively[55–57]. Thus, our findings in mice provide an anatomical framework for further exploring motor- and non-motor-functions of the cerebellum.

Our work provides four lines of evidence that strongly support a significant role for the cerebellum in coordinating striatal functions. First, our anatomical studies reveal that cerebello-striatal connections through the thalamus target not only medium spiny neurons (D1R-positive and D2R-positive) but provide major inputs to ChAT-positive interneurons in the dorsal striatum. Second, we report that DCN silencing results in a loss of the pS6rp activity marker in ChAT interneurons in the dorsomedial striatum. This highlights that endogenous activity of DCN outputs has functional impact on these striatal cells in vivo. Third, we demonstrate that acute chemogenetic silencing of 50–80% of IntP DCN neurons impairs the performance of mice in reward-driven alternation, a striatum-dependent task. Fourth, we demonstrate that striatal silencing of thalamo-striatal axons that arise from thalamic cells innervated by DCN axons replicates the behavioral phenotype in reward-driven alternation.

The activation of ChAT interneurons is thought to be critical for shifting behavioral strategies based on contextual information[29,41,42]. The phenotype in reward-driven alternation behavior observed with DCN silencing resembles phenotypes observed when striatal cholinergic function is disrupted[58–60]. Thus, our findings suggest a role for cerebellar outputs in this striatum-dependent behavior. Future studies should further delineate in detail the identity of thalamic relay neurons

contributing to disynaptic cerebello-striatal and cerebello-cortical connectivity, as well as the anatomical organization of the DCN outputs to these relays across deep cerebellar subnuclei. This anatomical and functional framework of cerebello-striatal connectivity will be essential for understanding how cerebellar computation impinges on striatal function and how cerebellar defects modify cognitive processes in disease states.

## Methods

**Mouse strains and ethical compliance.** All animal procedures were approved by the Cantonal Veterinary Office of Basel-Stadt, Switzerland and were performed in compliance with the Swiss laws. Transgenic mouse lines used in this study were: A2A-cre (Gensat, catalogue no. 031168-UCD), ChAT-cre (The Jackson Laboratory, Stock No. 018957), vGlut2-cre (The Jackson Laboratory, Stock No. 028863) D1R-cre (The Jackson Laboratory, Stock No. 028863), and Pcp2-cre[61]. All mice were maintained on C57Bl6J background except for A2A-cre and ChAT-cre which were on a mixed genetic background (FVB/C57Bl6J and 129S6/C57Bl6J, respectively). Strain- and sex-matched mice and littermates were used as negative controls. Anatomical experiments used male and female mice. Behavioral experiments were conducted exclusively with male mice.

**Viral vectors and virus production.** Attenuated rabies viruses for retrograde labeling (RV-ΔG-RFP = SADdeltaG-RFP[30] and monosynaptic tracing (RV$^{EnvA}$-ΔG-mCherry = EnvA-pseudotyped SADdeltaG-mcherry[36,62] were amplified and purified following published protocols. AAV2-CBh-GFP, AAV2-hSyn-DiO-hM4Di-mCherry, AAV8-EF1α-WGA-cre were purchased from UNC, vector core (University of North Carolina, USA). Adeno-associated viruses AAV2-hSyn-GFP, AAV2-EF1α-DiO-G-TVA (cre-dependent expression of rabies virus Glycoprotein and the TVA receptor under control of the EF1α promoter), AAV2-EF1α-DiO-G (cre-dependent expression of rabies virus Glycoprotein), and AAV2-hSyn-cre-GFP were generated as follows: Viral supernatants were produced by co-transfection of HEK293T cells (ATCC, verified by morphology and growth properties, potential mycoplasma contamination assessed every 6–12 months) grown on 15 cm dishes using calcium phosphate transfection of 70 μg of AAV helper plasmid (Rep/Cap, Serotype 2), 200 μg of AAV pHGTI-adeno1 (Plasmid factory) and 70 μg of AAV vector plasmid carrying the cDNAs to be expressed. 45–60 h after transfection, medium containing viral particles was harvested and purified using the Iodixanol purification method. Viral preparations were concentrated in Millipore Amicon 100 K columns at 4 °C. Virus samples were suspended in PBS, frozen in aliquots and stored at −80 °C (viral titers of the stocks used for injections were >10$^{11}$ particles/ml).

**Surgeries and stereotaxic injections.** Surgeries were performed under isoflurane anesthesia (Baxter AG, Vienna, Austria). The animals were placed in a stereotaxic frame (Kopf Instrument). Injections (200 nl/injection site) were made with a 33-G Hamilton needle (Hamilton, 65460-02).

In P18 mice the following stereotaxic coordinates were used for unilateral injection in tracing experiments (RV-ΔG-RFP and AAV2-hSyn-GFP tracing experiment):

*Interposed/medial deep cerebellar nucleus*: ML 1.33 mm, DV −3.76 and −3.64 mm from bregma, AP λ + 2.25 mm (2 injection sites).

*Dorsal striatum (contralateral to DCN sites)*: ML 1.8 mm, AP 0.2 mm, DV −3.3 mm and ML 2.0 mm, AP 0.0 mm, DV −3.2 mm from bregma (2 injection sites).

For intersectional marking of ILN neurons viruses were injected into the intralaminar nuclei and into the DCN contralateral to the striatal injection sites, of P11-P12 mice using stereotaxic coordinates:

*Intralaminar thalamic nuclei*: ML 0.6 mm, AP 1.6 mm, DV 3.5 mm and ML 0.9 mm, AP 2.0 mm, DV 3.8 mm from bregma (AAV2-DiO-mCherry, 2 injection sites).

*Interposed/medial deep cerebellar nucleus*: ML 1.25 mm, DV 3.55 mm from bregma, AP λ + 2.1 mm (AAV8-WGAcre).

For transsynaptic tracing from striatal cell populations defined by A2A-cre, ChAT-cre and D1R-cre expression, mice were first injected with at P11-P12 using the following coordinates:

*Dorsal striatum*: ML 1.8 mm, AP 0.5 mm, DV 3.8 mm from bregma (AAV2-EF1α-DiO-G-TVA).

*Intralaminar thalamic nuclei*: ML 0.6 mm, AP 1.6 mm, DV 3.5 mm and ML 0.9 mm, AP 2.0 mm, DV 3.8 mm from bregma (AAV2-EF1α-DiO-G-TVA and AAV2-hsy-cre-GFP co-injection).

One week later (P18–20), EnvA-pseudotyped rabies virus (RV$^{EnvA}$-ΔG-mCherry) was injected into the dorsal striatum: ML 1.8 mm, AP 0.2 mm, DV −3.3 mm and ML 2.0 mm, AP 0.0 mm, DV −3.2 mm from bregma (two injection sites).

We quantified GFP-positive striatal neurons in ChAT-cre mice ($N = 3$) to test if there was any undesired retrograde transport of AAV2-hSyn-cre-GFP from ILN to ST. We did not detect notable signal in these experiments.

For behavioral experiments, bilateral injections of purified AAV2-hSyn-DIO-hM4DGi-mCherry and AAV2-CBh-GFP were done at P18-P20, following stereotaxic coordinates:

*Interposed/medial deep cerebellar nucleus*: ML 1.33 mm, DV −3.76 and −3.64 mm from bregma, AP λ + 2.25 mm;

*Dorsal striatum*: ML 1.8 mm, AP 0.2 mm, DV −3.3 mm and ML 2.0 mm, AP 0.0 mm, DV −3.2 mm from bregma.

The virus was incubated for 2–3 weeks prior to performing the behavioral tasks or immunostaining. Injection sites in all animals were confirmed post hoc with brain sections.

In cre-negative mice, we did not detect any protein expression from cre-dependent (DiO) viral vectors. No viral infection was observed with pseudotyped RV$^{EnvA}$-ΔG-mCherry viruses in the absence of TVA expression. When cre-negative mice were injected with AAV2-EF1α-DiO-G-TVA and RV$^{EnvA}$-ΔG-mCherry, we did observe very sparse mCherry-positive cells in the dorsal striatum ($n = 21$ cells on average) and intralaminar thalamic nuclei ($n = 3$ cells on average) but no cells in the DCN or cerebellum. This indicates a very low level leakage of TVA expression and consequent RV$^{EnvA}$-ΔG-mCherry infection in the absence of transsynaptic spread—presumably due to limited glycoprotein expression. Thus, the retrograde tracing of cerebello-thalamic connections is not significantly affected by this spurious infection.

**Generation of anti-rabies glycoprotein antibody**. As antigens, two peptides were selected from the rabies virus glycoprotein sequences and synthesized: C+ISS-WESHKSGGETRL and TTTFKRKHFRPTPDAC (single amino acid code, N to C-terminus, C + indicates a cysteine added to the N-terminus of one of the peptides for thiol-mediated coupling). The synthetic peptides were conjugated to keyhole limpet hemocyanine for immunization of two rabbits and two guinea pigs (Eurogentec, Belgium). Initial immunization was followed by three boosts (2–4 week intervals) and animals were exsanguinated. All sera recognized rabies virus glycoprotein expressed in HEK293T cells. Antibodies were affinity purified on the peptide antigens coupled to sepharose beads, and eluted with glycine-HCL, pH2.5. Affinity-purified guinea pig antibodies were used for immunohistochemistry on rabies glycoprotein-expressing brain sections, as they performed better than the antibodies produced in rabbits.

**Cannulation and local CNO infusion**. Bilateral cannulae (26 GA, Plastics One) were placed at ML 2.05 mm, AP 0.0 mm, DV −2.45 mm from bregma, so the cannula openings locate in the middle of dorsal striatum in terms of ML and AP axis. Dental cement (G-ænial) was used to anchor the guide cannula to the skull. Dummy cannula (Plastics One) were inserted to keep the fiber guide from getting clogged. After surgery, mice were allowed 5–7 days for recovery. Bilateral intra-cranial injections of 30 μM CNO (800 nl/side) were infused with a microinjection pump (Harvard apparatus) using a speed of 160 nl/min. When behavior tests were finished and before perfusion, 800 nl of 2 mg/ml dextran Alexa 488 (sigma) diluted in saline were cranially infused into each cannula for the post hoc histology. In pilot experiments, 200 nl, 400 nl and 800 nl of 2 mg/ml dextran Alexa 488 were infused, and 800 nl volume was found to have best coverage of striatum without leakage to surrounding brain areas. After perfusion, we found $n = 4$ animals from hM4Di group and $n = 3$ animals from the tdTomato group did not have a correct cannulation positions, therefore these mice were excluded from the dataset.

**Immunohistochemistry**. Mice were deeply anesthetized and transcardially perfused with PBS 1 × followed by 4% paraformaldehyde prepared in PBS. The brain was removed and left for post-fixation at 4 °C in PBS. Horizontal brain slices (with the caudal side slightly tilted towards dorsal) were cut at 50 μm with a vibratome (Leica Microsystems VT1000, Germany). For immunohistochemistry, brain sections were kept in PBS before incubation with blocking solution containing 0.5% Triton X-100 in Tris-buffered saline and 10% normal donkey serum. Slices were incubated with primary antibodies at room temperature overnight and washed three times in PBS containing 0.5% Triton X-100, followed by incubation for 2 h at room temperature with a secondary antibody. Sections were washed three times in PBS before mounting onto microscope slides with Fluoromount-G (SouthernBiotech, 0100-01).

The following primary antibodies were used in this study: guinea pig anti-rabies G (described above, 1:500); guinea pig anti-vGluT2 (Synaptic Systems, 135404, 1:500); guinea pig anti-vGAT (Synaptic Systems, sep-62, 1:1000); sheep anti-TH (Millipore, AB1542, 1:1000); rabbit anti-PENK (Labforce, LS-C23084, 1:500); rabbit anti-ChAT (Millipore, AB143, 1:100). Mouse anti-NeuN (Abcam, ab17787, 1:500), polyclonal rabbit anti-p-Ser$^{240-244}$-S6rp (1:200, #2215, Cell Signaling Technology, Beverly, MA) and polyclonal goat anti-ChAT (1:100, #AB144P, Millipore, Billerica, MA). Secondary antibodies were: donkey anti-guinea pig IgG-Cy5 (Jackson ImmunoResearch, 706-175-148, 1:1000), donkey anti-sheep IgG-Cy5 (Jackson ImmunoResearch, 713-165-147, 1:1000), goat anti-rabbit IgG-Alexa 488 (Abcam, 1:500; ab150077), goat anti-mouse IgG-Alexa 488 (Abcam, 1:500, ab150113). Images were acquired with a LSM-700 confocal microscope or Zeiss AxioScan.Z1 Slidescanner. Given that immunological reagents to selectively mark D1R-positive MSNs are limiting, we could not examine this cell population in the WGA transneuronal tracing experiments.

**Quantitative analysis**. For examining the DCN-derived axons and ILN neurons back-labeled by RV-ΔG-RFP, as well as ILN-derived axons and striatal neurons,

confocal stacks were taken on a Zeiss LSM700 confocal microscope (25×, NA 0.8 and 40×, NA 1.3 objectives). For the analysis of cortical and striatal coverage by ILN-derived axons, images stacks were carried out with Fiji after selecting 400 μm x 400 μm fields in each cortical area and in the dorsal striatum. Images were first thresholded in Fiji to generate binary images, then analyzed by Fiji using analyze particles functions. For mapping the percentage of GFP and mCherry positive neurons in multisynaptic tracing experiment, every second brain section was sequentially mounted and imaged by Zeiss AxioScan. Z1. Then the GFP and mCherry positive neurons were manually counted in those images. For mapping the location of DCN neurons and Purkinje cells, brain sections were manually registered to the mouse brain atlas (Paxinos and Franklin's, fourth edition) with DCN subnuclei or shape of cerebellar lobules using the Fiji landmark transformation function.

To assess the percentage of DCN neurons expressing hM4Di in the behavioral experiments in Fig. 6, every second 50 μm section containing DCN substructures was stained with anti-NeuN antibodies and images were acquired on a Zeiss AxioScan.Z1 Slidescanner. The NeuN- and mCherry-positive cells in the DCN substructures were counted by an investigator blinded to the animal ID in Fiji. The percentage of hM4Di-mCherry-expressing neurons was calculated for each animal. In Fig. 6c–i only behavioral data from mice with >50% interposed neurons expressing hM4Di-mCherry were included for the behavioral phenotype assessment. In Fig. 6j the behavioral performance of all injected animals (no cut-off for percentage of infected cells) was correlated with the hM4Di-expression level in DCN subnuclei.

For the quantitative analysis of pS6rp staining in dorsal striatum ChAT neurons, DCN$^{hM4Di}$ or control mice were i.p. injected with 2.5 mg/kg CNO and maintained in their home cage. 50 min later, animals were anesthetized and transcardially perfused. Sections (50 μm thick, spaced 150 μm apart) from the dorsal striatum were immunostained with anti-ChAT and anti-pS6rp antibodies and images of ChAT interneurons were captured from medial and lateral striatum on a Zeiss LSM700 confocal microscope (63× objective). Focal planes with maximal ChAT immunostaining signal were acquired with identical laser power and photomultiplier settings by an investigator blinded to virus infection rate in the DCN. Raw 8-bit images were analyzed with Fiji. The fluorescence intensity of the pS6rp signals in the soma of each ChAT neuron was determined using the ChAT signal as a mask (subtracting the nuclear area). Local background in the vicinity of the cell soma was measured and subtracted from the somatic signal. A pseudo-color palette (16-color LookUp table) was used to display signal intensity.

In the intersectional viral manipulation experiment in Fig. 7, AAVs injected into DCN were using high titer preparations and consistently infected large fractions of DCN neurons. Thus, animals were not grouped according to the efficiency of viral DCN infection (as for the experiments in Fig. 6c–i were overall infection efficiency was lower) and all animals with correctly placed cannulae in the striatum were included in the quantitative analysis.

**Electrophysiology**. For acute slice recording, postnatal day 30–40 mice, injected at P18 with AAV2-DIO-ChR2-GFP and AAV2-cre-venus in DCN were anesthetized with isoflurane and decapitated. Three hundred micrometer thick horizontal sections were cut in sucrose substituted artificial cerebrospinal fluid (ACSF) that consisted of 87 mM NaCl, 2.5 mM KCl, 1.25 mM NaH2PO4, 25 mM NaHCO3, 25 mM glucose, 75 mM sucrose, 0.5 mM CaCl2, 7 mM MgCl2. Slices were allowed to recover at 34 °C for 1 h and then maintained at room temperature in the same sucrose ACSF. For whole-cell recordings, slices were perfused with 125 mM NaCl, 2.5 mM KCl, 1.25 mM NaH2PO4, 25 mM NaHCO3, 2 mM CaCl2, 1 mM MgCl2, 25 mM glucose, 50 uM picrotoxin. For all experiments, whole-cell recordings were digitized at 10 kHz and filtered at 2 kHz. Whole-cell patch-clamp recordings of ILN neurons were made with an EPC10/double patch-clamp amplifier (HEKA) under visualization in an upright microscope (Olympus) equipped with gradient contrast infrared visualization (Luigs and Neumann) and a 60× objective. Patch pipettes had resistance of 2.7–3.5 MΩ and were filled with an internal solution that contained 120 mM Cs-gluconate, 40 mM Hepes, 4 mM NaCl, 2 mM QX314, 2 mM Mg-ATP, 0.3 mM NaGTP, 0.3 mM EGTA, 305 mOsm, pH 7.2. The cells were held at a holding potential of −70 mV, and +40 mV when measuring NMDA current.

DCN-derived axons expressing ChR2 were photostimulated with a monochromater (Polychrome V, FEI) coupled into the epifluorescence port of the microscope, with wavelength of 470 nm. Light was triggered by HEKA. Light power was measured as 3.865 mW/mm$^2$ through an Olympus 10× objective (air, NA 0.3). Responses were first assessed with 20 ms light pulses. To assess NMDAR/AMPAR ratios 100 ms light pulses were applied.

**Behavioral experiments**. Mice were habituated to the experimenter by daily handling for 1 week before the behavioral experiments. On test day, hM4Di-expressing and GFP-expressing mice were i.p. injected with 2.5 mg/kg CNO 30 min before the test.

For food restriction, mice were deprived of food for the first 24 h, then was given 1.5–2 g food pellet per mouse per day. Mouse body weight was maintained at 80–90% of the weight with food provided ad libidum. Access to drinking water was always unrestricted. One with GFP injected mice died before the food restriction started.

For open field assays, mice were placed in an arena (50 × 50 × 25 cm) under normal room light for 7 min. Movement in the open field was recorded with a BASLER Ethovision Camera (Noldus). Total distance traveled and velocity were quantified using Ethovision 10 software (Noldus). Velocity was calculated by dividing distance traveled by moving time, i.e., not including pause periods where the mouse was immobile.

For marble burying assays, mice were placed in a standard Type II cage with 5 cm bedding containing 20 identical, evenly distributed black marbles for 30 min. A marble was considered buried at the end of the test if at least 2/3 of the marble was covered with bedding.

For T-maze spontaneous alternation assays, mice were tested in an elevated T-maze with a start compartment leading to one open arm and two closed target arms (35 cm × 6 cm and 74 cm above the ground)[49]. Each mouse performed 10 trials. In each trial, mice were placed at the start position and a 1 kHz tone (10 s) was played to signal the start. After a 10 s waiting period, the door was opened and mice could move along the open arm and choose to enter either one of the two closed arms. An arm was considered chosen when the whole body and tail of the mouse had entered the arm. After making a choice, a separation wall was inserted and the mouse was confined to the chosen arm for 30 s and then placed in the start compartment for the next trial. Runs were scored as an alternation when the mouse chose the opposite arm from the arm selected in the previous trial. Trials were aborted when mice did not leave the start area within 60 s of the auditory start signal or did not enter into an arm after 90 s.

After completing the spontaneous alternation test, mice were food-restricted for 7–10 days and tested in T-maze reversal learning:[48,58]. Mice were trained in the T-maze in 10 trials per day ( = one session per day) to retrieve food from the end of one arm of the T-maze. Food was placed in the arm opposite to the one the mice had preferred in the spontaneous alternation experiment. This same arm was then used throughout the remaining trials. Start was signaled with a 1 kHz tone (10 s), arm choice criterion and criteria to abort a trial were as for spontaneous alteration. Once the mouse chose an arm it was confined to the target area for 30 s to consume the food pellet (on the rewarded side) or to explore that area (in case the unrewarded side had been chosen). After reaching the learning criterion (80% correct choices in one session), reversal learning was tested on the following 2 days (one session per day). The food pellet was placed on the side opposite to the trained side. Thirty minutes before the start of each session, animals were i.p. injected with 2.5 mg/kg CNO. The rate of correct/incorrect choices was recoded. Mice that did not reach the learning criterion after 10 session were excluded from the analysis (this was the case for one GFP-expressing and two hM4Di-expressing mice).

After completion of the T-maze reversal learning mice were maintained for 4–7 days under food restriction and tested in T-maze forced alternation[49]. Mice were trained 11 trials per day ( = one session per day) in the T-maze in a forced alternation task. For the first trial of each session, food pellets were placed on both sides of the closed arm and the mouse could choose either arm to obtain the food. From the second trial onwards, food pellets were placed in the arm opposite the arm visited in the previous trial, regardless of whether the mouse received the food pellet or not. After reaching the learning criterion (70% correct choices in the two sequential sessions), mice were i.p. injected with 2.5 mg/kg CNO on the following day and tested for the forced alternation behavior 30 min after injection. Mice that did not reach the learning criterion after 12 session were excluded from the analysis (this was the case for one hM4Di-expressing mouse).

**General statistical methods**. Sample sizes were determined based on previous experience with the manipulations, advice from colleagues, and literature surveys. A formal sample size calculation was not performed as the effect sizes were unknown during the time of study design.

The following preestablished exclusion criteria in the animal experiments were defined to ensure successful implementation of the experimental manipulation and/or to avoid effects to unspecific confounding factors: For stereotaxic injection experiments, appropriate center of the injection sites was confirmed in post hoc anatomical analyses. All animals with mis-targeted injections were excluded from the analysis. The same applied for the positioning of cannulae for local CNO infusion. A further preestablished criterion was exclusion of animals that differed by 20% in weight from the mean of the group or that exhibit visible behavioral abnormalities.

Animals were randomly assigned to groups injected with control viruses or for chemogenetic manipulations.

Investigators performing behavioral scoring and anatomical analysis (e.g. of percentage of infected cells or viral targeting specificity) were blinded to the group allocation and/or previously collected behavioral data.

Appropriate statistical tests were chosen based on sample size, normality of distribution of data points and number of groups to be compared. Variance of the groups being compared was similar. Details on n numbers, p-values and specific tests are noted in the figure legends.

**Data availability statement**. Additional data that support the findings of this study and that could not be included in the manuscript due to space restrictions are available from the corresponding author upon reasonable request.

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

## Acknowledgements

We thank members of the Scheiffele Lab for advice and illuminating feedback. We thank Lisa Traunmüller, Oriane Mauger, Elisabetta Furlanis, Yunyun Han for constructive discussions and comments on the manuscript. We are grateful to Tev Stachniak and Kelly Tan for expert advice on cannulation and local CNO-mediated silencing and Dheeraj Malhotra for providing advice on statistics. We thank the Arber Laboratory at the University of Basel for sharing viral tracing reagents and the Imaging Core Facility of the University of Basel, in particular Kai Schleicher and Niko Ehrenfeuchter. This work was supported by funds to P.S. from the Swiss National Science Foundation, the NCCR SYNAPSY, EU-AIMS which receives support from the Innovative Medicines Initiative Joint Undertaking., and the Kanton Basel-Stadt.

## Author contributions

Stereotaxic injections, histology, and quantitative analysis of anatomical experiments reported in Figs. 1–7 and Supplementary Figs. 1, 2, 3, and 5 were done by L.X. and C.B. Physiological recordings in Supplementary Fig. 1, behavioral experiments in Fig. 6 and Supplementary Figs. 6 and 7, and statistical analyses for all data were done by L.X. Cannulations and behavioral experiments reported in Fig. 7 were done by L.H-B. and L.X. Preparation of viral vectors and stocks, and antibody production reported in Supplementary Fig. 4 were done by C.B. The study was designed and the manuscript written by L.X. and P.S.

## Additional information

**Competing interests:** The authors declare no competing interests.

