## [Peer Review File · Nature Communications]

Reviewers' comments:

Reviewer #1 (Remarks to the Author):

Previous works have shown that the deep cerebellar nucleus could communicate with the dorsal striatum via thalamic intralaminar nucleus (ILN). In the present study, Xiao et al. (Scheiffele lab) extended the analysis on the extend, and cell-types involved in this di-synaptic communication. They used simple and straight-forward experimental design: to use AAV2-GFP to label DCN, and to examine contacts of their axons with neurons retrogradely marked from dorsal striatum using rabies virus (RV - dG-RFP). This analysis convincingly demonstrated that vGluT2-positive DCN neurons form synaptic contact with ILN neurons that were retrogradely labeled from the dorsal striatum. DCN neurons project to contralateral but not ipsilateral ILN neurons. To further dissect the connection, they showed that thalamic axons could D1, D2 MSNs, and ChAT interneurons. Lastly, to demonstrate the behavioral relevance, the authors used chemogenetic tools to selectively silence IntP DCN neurons and found that such manipulation could impact ChAT interneuron activity, and a reward-driven striatum-dependent behavior task (but not normal locomotion). Overall, the work is nicely done, and data are presented clearly, and are of high quality.

1. The major weakness is that many main conclusions are supporting previous findings. For example, a) the di-synaptic connection between DCN and the dorsal striatum through intralaminar thalamic nucleus; b) thalamic neurons form synaptic connection with D1, D2 MSNs, and ChAT. However, current work is done using the currently widely-used, improved rabies virus labeling. The quality of the data is high. Overall, the current study could provide further information over previous findings.
2. One of the novel findings is that silencing the IntP DCN could impact neuronal activity markers in dorsomedial striatal ChAT interneurons. This is an interesting finding. However, a few questions are not answered. Do silencing DCNs only impact ChAT interneurons? How about the neighboring MSNs? What is the explanation for different pS6rp immunoreactivity seen in dorsomedial and dorsolateral ChAT interneurons? Only zoomed in representative neurons are shown in Fig6. The authors should also demonstrate lower magnification images to give the audience a better understanding on overall staining in the dorsal striatum.
3. The correlation between the fraction of hM4Di-mCherry positive neurons and the pS6rp levels in the striatum is interesting. However, the conclusion made through this set of experiments needs to be carefully re-considered. It is not clearly stated what is the cause of different levels of hM4Di-mCherry? Do authors argue different levels of synaptic connection? Or possibly different levels of viral infection? When calculate the correlation, are data from wt animals included to establish better baseline level?
4. Based on the anatomy data, one would assume that silencing DCN would reduce contralateral ILN neuronal activity. Does silencing ILN activity could also lead to change in pS6rp levels in ChATs? Would the effect of silencing DCN be blocked or reversed by enhancing ILN activity?
5. Another novel finding is the behavioral study. It is very impressive, and the interpretation is interesting. One shortcoming is that, in the behavioral study, the silenced DCN neurons are not specific to those who make di-synaptic connections with the dorsal striatum. In addition, to make the overall logic sound, the study could be improved by manipulating thalamic ILN neurons in conjunction of silencing DCN neurons. Technically, this is doable since thalamic neurons are also vGluT2. For example, would the behavioral effect of silencing DCN be blocked or reversed by enhancing ILN activity? Would the behavioral effect of silencing DCN be modulated by manipulating dorsomedial (or dorsolateral) ChAT interneuron activity?

Reviewer #2 (Remarks to the Author):

Major comments:

The authors present interesting findings that DREADD-mediated silencing of deep cerebellar nuclei (DCN) reduces activity of ChAT+ striatal interneurons and has a deleterious effect on a forced-alternation task. However, the "disynaptic" transsynaptic tracing using rabies virus, which comprises a great deal of the paper (figures 3-5), is flawed.

The authors attempt to achieve disynaptic tracing by using the usual monosynaptic tracing system in three different Cre lines (D1R-, A2A- and ChAT-Cre) to allow rabies virus to spread to the (intralaminar nuclei of the) thalamus from these starting cell groups, in mice that they have also injected AAV nonspecifically expressing rabies virus glycoprotein in the ILN. This is meant to allow spread of the rabies virus an additional synaptic step to the DCN neurons that are presynaptic to the ILN cells that are presynaptic to the three striatal populations.

The fundamental problem with this approach is that nonspecific expression of G in the ILN (or probably more broadly in the thalamus, because it would be difficult to restrict an injection to the ILN, and we can't really see clear GFP signal in the images) means that G is expressed not only in any thalamic neurons that are presynaptic to the starting cells in striatum, as intended, but also in any thalamic neurons that are presynaptic to those directly-presynaptic thalamic neurons, and so on. So then the virus would spread through those indirectly-connected thalamic cells and label *their* inputs, as well as the inputs to the actually monosynaptically connected ILN cells.

This approach (which I know has been used by others) would only make sense if there were no connections between thalamic neurons. In that case, the channels would indeed be separate.

Similarly, the "trisynaptic" version (Figure 5) also requires that there are no connections between DCN neurons to be valid.

Also, the authors do not show the results of the much simpler monosynaptic tracing experiment that is the precursor of the "disynaptic" one, in order to see where in the ILN/etc. the different thalamic inputs to the three striatal populations are. Understood that getting at the disynaptic, rather than monosynaptic, inputs is the point of the paper, but showing the results of the simpler experiment would help in interpretation of the more complex ones.

A possible approach at this point might be, having developed hypotheses about the locations in DCN of the neurons disynaptically connected to the striatal cell groups, to simply use monosynaptic tracing to label their direct inputs in ILN, in combination with injection of AAV-ChR2 in DCN, then do whole-cell recordings ex vivo with optogenetic activation of the (cut) DCN axons to see whether a given region of DCN indeed connects to the cells that are presynaptic to a given striatal cell type.

Re detection of the GFP signal to see where G is really expressed, the best thing for tracking Cre expression would have been to do the experiments in Cre reporter mice (Ai14, etc., crossed to the Cre lines, of course), which would allow sensitive detection of where Cre was expressed at high enough levels to have a functional effect; second best would have been to immunostain for GFP) in order to see what cells are expressing G (EGFP-Cre being a proxy for G expression, given that immunostaining for G is evidently an unsolved problem in the field). In a similar vein, "We quantified GFP-positive striatal neurons in ChAT-cre mice (N=3) to test if there was any undesired retrograde transport of AAV2-hSyn-cre-GFP from ILN to ST. We did not detect notable signal in these experiments" doesn't mean much, because we can't see the GFP anyway, and Cre would be effective at a very low level. This is a relatively minor point, because I would not expect a great deal of retrograde transport of AAV2.

Very minor comments:

p.1: "chemigenetics" -> "chemogenetics", more common term and that used by Roth.

p.2, line 35: "up-date" -> "update"

p.6+: "mono-synaptic" -> "monosynaptic"; "trans-neuronal" -> "transneuronal", "trans-synaptic" -> "transsynaptic" to follow most common forms of these words

Reviewer #3 (Remarks to the Author):

The manuscript by Xiao and colleagues shows disynaptic anatomical connection between the cerebellum and the striatum, through the thalamus. They show that this connection reaches D1 and D2 MSNs, and also very strongly ChAT-positive interneurons. The authors show that if they silence DCN with DREADDs, activity-dependent gene expression decreases significantly in ChAT neurons in DMS. However, the authors do not show that this effect is mediated through the thalamus. The main issue of this study is in the last figure - behavior. The effects are weak, and the exclusion/division criteria applied of 50% is arbitrary. The authors should show an overall correlation (not only of animals with more than 50% infection, before applying any exclusion criteria). Also, the authors claim that the T-maze alternation task is striatal-dependent, but that is not clear. Finally, it would be important to show that the behavioral effects depend indeed on the disynaptic connection from DCN to striatum via thalamus.

So overall, the proof of disynaptic anatomical connection from DCN to striatum is strong, but the functional characterization of this disynaptic connection from a behavioral standpoint is weak and should be improved. As it stands, the behavioral conclusions are not supported.

Minor points:

- In the confocal images, the authors should show overlap using z-stack.
- Figure 6e/f panels seem to be reversed
- The authors do not seem to normalize the cell counts for the number of starter cells
- Methods are missing for quantification of cell counts for Fig 2, 3
- Fig 2h, they should show D1 stainings as well, only showing D2 and ChAT (in supp?)

Stats/uncertainties:

- Figures 3e, 4e, 7g, error bars not explained

Reviewer #4 (Remarks to the Author):

This manuscript explores the organization and function of the di-synaptic connection from the deep cerebellar nuclei (DCN) to the striatum in mice. Understanding how cerebellar outputs influence striatal activity is highly relevant to the field of neuroscience because of wide implications for motor and cognitive functions, as well as for multiple disorders. As discussed below, the authors claim to provide three independent lines of evidence for cerebellar influence on striatal activity, via the intralaminar nuclei (ILN) of the thalamus.

1. The authors use anatomical tracing techniques and identify di-synaptic projections from the DCN to multiple classes of neurons in the striatum. The anatomical results are not entirely novel, as the authors imply. However, they do expand on the understanding of the di-synaptic projection from the DCN to the striatum.

First, previous findings in primates have shown that inputs to the striatum originate from all of the DCN (Hoshi et al., 2005). Thus, the fact that all the DCN send di-synaptic projections to the striatum is not novel and thus, the authors' focus on the medial and interposed nuclei is limiting. This limitation raises the question of how projections from these nuclei compare with projections from the lateral nucleus. This question is particularly relevant given that the lateral nucleus is the source of a majority of cerebellar output.

Second, studies in mice have shown that stimulation of the DCN modulate a variety of neuronal types in the striatum, including medium spiny neurons (MSNs), fast spiking interneurons, and potentially also cholinergic (ChAT) interneurons (Chen et al., 2014). Using cell-type specific tracing techniques, the authors show that all of the DCN send di-synaptic projections to MSNs that express both D1 and D2 receptors. This aspect of the results is new and interesting; it suggests that both the direct and indirect pathways through the striatum may be influenced by cerebellar activity in mice (see Bostan & Strick 2018 for a discussion of potential selectivity of cerebellar outputs to the indirect pathway).

The most interesting finding in this manuscript is arguably the direct evidence for DCN di-synaptic projections to ChAT interneurons in the striatum. As the authors appear to agree, this finding encourages further exploration. Arguably, focus on the organization and function of the di-synaptic projection from the DCN to ChAT interneurons would provide a more cohesive theme for the manuscript. Although the authors do pursue related questions, their results provide limited insights into the functions of this pathway (see points 2 and 3).

Another interesting result is that individual DCN send differential projections to specific cell types in the striatum (Figure 4). As the di-synaptic projections from all the DCN reach all the striatal cell types, the widespread location of Purkinje cells that project tri-synaptically to D2 MSNs (Figure 5) is not particularly surprising. Although the authors do not explore this issue, Figure 4 suggests some level of organization of di-synaptic projections to specific cell types in the striatum within individual nuclei. It would be useful if the authors provided insights into this organization. Furthermore, it would be informative to determine whether multi-synaptic projections from Purkinje cells to specific cell types in the striatum are (also) topographically organized. Such organization might provide some insights into the potential diversity of functions of the cerebello-striatal connection. Thus, it is unclear why the authors chose to only study the multi-synaptic projection from Purkinje cells to the striatum in the A2A-cre mice.

2. The authors use selective silencing of deep cerebellar nuclei to assess their effects on striatal ChAT interneuron activity. Previous work has shown interesting effects on markers of ChAT interneuron activity and related behaviors after lesions of ILN (e.g., Bradfield et al. 2013). As expected from this previous work, the authors show differential effects of DCN neuron silencing on markers of ChAT interneuron activity in the dorsomedial striatum (DMS) vs. dorsolateral striatum (DLS). However, because of the unconstrained injection sites in the anatomical studies performed here, the reason for differential effects of DCN on DMS vs. DLS remains unclear. Determining how individual DCN target specific cell types (including ChAT interneurons) in functionally different striatal regions would go a long way in clarifying the different functions of this pathway.

3. The authors investigate the behavioral effects of selective silencing of deep cerebellar nuclei on behavior. This is the least well-developed aspect of the paper and not particularly informative. First, the authors do not provide electrophysiological that the DCN have actually been silenced. Second, there is no behavioral evidence for functional silencing of the DCN. It is unclear why the mice were not evaluated on a task that is dependent on DCN function. Finally, the authors test the effects of silencing the DCN on "striatum-dependent" tasks and note that performance is reminiscent of mice with abnormal ChAT interneuron function. The authors' interpretation of this data is a stretch for two main reasons: 1) the authors do not use tasks that have been shown to be ChAT interneuron dependent (e.g., Bradfield et al. 2013) in order to properly compare their results with previous findings, and 2) although the "forced alternation" T-maze task involves mice obtaining a reward, there are multiple possible routes for DCN outputs to influence behavior on this task that do not involve DCN projections to the striatum (e.g., projections to cortex).

References:

Bostan, A. C. & Strick, P.L. The basal ganglia and the cerebellum: nodes in an integrated network.

Nature Reviews Neuroscience, doi: 10.1038/s41583-018-0002-7 (2018)

Bradfield, L. A., Bertran-Gonzalez, J., Chieng, B. & Balleine, B. W. The thalamostriatal pathway and cholinergic control of goal-directed action: interlacing new with existing learning in the striatum.

Neuron 79, 153-166, doi:10.1016/j.neuron.2013.04.039 (2013).

Chen, C. H., Fremont, R., Arteaga-Bracho, E. E. & Khodakhah, K. Short latency cerebellar modulation of the basal ganglia. Nat Neurosci 17, 1767-1775, doi:10.1038/nn.3868 (2014).

Hoshi, E., Tremblay, L., Feger, J., Carras, P. L. & Strick, P. L. The cerebellum communicates with the basal ganglia. Nat Neurosci 8, 1491-1493, doi:10.1038/nn1544 (2005).

We thank all four reviewers for their thoughtful assessment of this work. We have marked the sections with major text edits in the manuscript in blue font. Below is a point-by-point response to all reviewer comments (answers in blue).

Reviewer #1 (Remarks to the Author):

Previous works have shown that the deep cerebellar nucleus could communicate with the dorsal striatum via thalamic intralaminar nucleus (ILN). In the present study, Xiao et al. (Scheiffele lab) extended the analysis on the extent, and cell-types involved in this di-synaptic communication. They used simple and straight-forward experimental design: to use AAV2-GFP to label DCN, and to examine contacts of their axons with neurons retrogradely marked from dorsal striatum using rabies virus (RV-dG-RFP). This analysis convincingly demonstrated that vGluT2-positive DCN neurons form synaptic contact with ILN neurons that were retrogradely labeled from the dorsal striatum. DCN neurons project to contralateral but not ipsilateral ILN neurons. To further dissect the connection, they showed that thalamic axons could D1, D2 MSNs, and ChAT interneurons. Lastly, to demonstrate the behavioral relevance, the authors used chemogenetic tools to selectively silence IntP DCN neurons and found that such manipulation could impact ChAT interneuron activity, and a reward-driven striatum-dependent behavior task (but not normal locomotion). Overall, the work is nicely done, and data are presented clearly, and are of high quality.

We thank the reviewer for the careful evaluation of the work and for highlighting the strengths of the paper.

1. The major weakness is that many main conclusions are supporting previous findings. For example, a) the di-synaptic connection between DCN and the dorsal striatum through intralaminar thalamic nucleus; b) thalamic neurons form synaptic connection with D1, D2 MSNs, and ChAT. However, current work is done using the currently widely-used, improved rabies virus labeling. The quality of the data is high. Overall, the current study could provide further information over previous findings.

We thank the reviewer for highlighting the quality of the data in our manuscript and the advanced methods employed to deepen our understanding of cerebello-striatal connectivity. Vis-à-vis the previous work we feel that there are three major findings that make our work more than just an incremental extension of the important previous findings:

- (1) We demonstrate that cerebellar outputs are anatomically as well as functionally related to ChAT interneurons in the striatum.
- (2) We demonstrate that silencing of cerebellar outputs impacts behavior in a striatum-dependent behavioral task.
- (3) We demonstrate that local silencing of thalamo-striatal axons that arise from cells innervated from the DCN is sufficient to impair the striatum-dependent behavior (new data in Figure 7).

We consider the much more fine-grained anatomical dissection as well as the demonstration of a functional relevance of this pathway for a striatal behavior a major advance.

2. One of the novel findings is that silencing the IntP DCN could impact neuronal activity markers in dorsomedial striatal ChAT interneurons. This is an interesting finding. However, a few questions are not answered. Do silencing DCNs only impact

ChAT interneurons? How about the neighboring MSNs? What is the explanation for different pS6rp immunoreactivity seen in dorsomedial and dorsolateral ChAT interneurons? Only zoomed in representative neurons are shown in Fig6. The authors should also demonstrate lower magnification images to give the audience a better understanding on overall staining in the dorsal striatum.

A selective impact of thalamo-striatal connections on ChAT interneurons in dorsomedial striatum was first reported by Balleine and colleagues in rats (Neuron, 2013). However, their study (or any other study) never examined whether endogenous activity of DCN outputs can modify the state of ChAT interneurons. We note that the baseline ChAT interneuron p-S6rp immunoreactivity is lower in dorsolateral as compared to dorsomedial striatum, indicating that their activity and/or signaling state is different. This might be the reason why we do not observe a further reduction upon DCN silencing. We now discuss this in the revised manuscript.

We have now provided the lower magnification views of pS6rp staining as requested by the reviewer (**new Figure 5a,b**). These new images show that p-S6rp immunoreactivity is also detected in some ChAT-negative cells. However, it is currently unknown whether these cells are MSNs and how neuronal activity might regulate pS6rp in MSNs. Further work will be required to conclude whether it is a suitable marker for these cells (just as c-Fos is not up-regulated in all classes of neurons in response to neuronal activity). We clarified this in the revised manuscript.

3. The correlation between the fraction of hM4Di-mCherry positive neurons and the pS6rp levels in the striatum is interesting. However, the conclusion made through this set of experiments needs to be carefully re-considered. It is not clearly stated what is the cause of different levels of hM4Di-mCherry? Do authors argue different levels of synaptic connection? Or possibly different levels of viral infection? When calculate the correlation, are data from wt animals included to establish better baseline level?

We apologize for not having explained this point with sufficient clarity in the original manuscript. The different “levels” of hM4Di-positive neurons in the DCN are a consequence of differential efficiency of viral infection obtained with the stereotaxic injections. We now specified this in the text. We did indeed use 3 uninfected wild-type animals that lack hM4Di-positive cells as a baseline (data points for 0% hM4Di-positive neurons in Figure 5f and 5j). We now clearly specified this in the figure legend.

4. Based on the anatomy data, one would assume that silencing DCN would reduce contralateral ILN neuronal activity. Does silencing ILN activity could also lead to change in pS6rp levels in ChATs? Would the effect of silencing DCN be blocked or reversed by enhancing ILN activity?

5. Another novel finding is the behavioral study. It is very impressive, and the interpretation is interesting. One shortcoming is that, in the behavioral study, the silenced DCN neurons are not specific to those who make di-synaptic connections with the dorsal striatum. In addition, to make the overall logic sound, the study could be improved by manipulating thalamic ILN neurons in conjunction of silencing DCN neurons. Technically, this is doable since thalamic neurons are also vGluT2. For example, would the behavioral effect of silencing DCN be blocked or reversed by enhancing ILN activity? Would the behavioral effect of silencing DCN be modulated by manipulating dorsomedial (or dorsolateral) ChAT interneuron activity?

We thank the reviewer for these suggestions that are aimed to strengthen the link

between DCN outputs, the intralaminar nucleus and striatal cells and function. We considered various potential experiments to address this. Instead of activating ILN neurons under conditions where DCN neurons are silenced (as suggested by the reviewer), we decided to selectively silence the activity of striatal projections from the ILN cells that are innervated from the DCN. We think that this loss-of-function experiment is better suited to probe the importance of cerebello-striatal connectivity for the behavior than a gain-of-function in ILN (which may not replace the information resulting from cerebellar computation and DNC outputs).

Thus, we used an intersectional viral strategy to drive hM4Di in thalamic cells that are contacted by DCN axons (expressing WGA-cre – as previously established in the anatomical experiments for Figure 2). We then cannulated mice and locally infused CNO into the dorsal striatum. This enabled us to test whether the silencing of the ILN-derived axons that relay DCN outputs would impair performance of the mice in the T-maze task. We found that this striatal silencing indeed recapitulated the behavioral phenotype observed for silencing of DCN neurons (**new Figure 7**). We think that this experiment provides strong evidence that the behavioral phenotype indeed results from silencing of DCN outputs routed via the thalamus to the striatum, rather than an action of DCN outputs via thalamocortical projections.

Reviewer #2 (Remarks to the Author):

Major comments:

The authors present interesting findings that DREADD-mediated silencing of deep cerebellar nuclei (DCN) reduces activity of ChAT+ striatal interneurons and has a deleterious effect on a forced-alternation task.

We thank the reviewer for his/her appreciation of the data on the activity of ChAT-positive striatal interneurons and the behavioral read-outs.

However, the "disynaptic" transsynaptic tracing using rabies virus, which comprises a great deal of the paper (figures 3-5), is flawed. The authors attempt to achieve disynaptic tracing by using the usual monosynaptic tracing system in three different Cre lines (D1R-, A2A- and ChAT-Cre) to allow rabies virus to spread to the (intralaminar nuclei of the) thalamus from these starting cell groups, in mice that they have also injected AAV nonspecifically expressing rabies virus glycoprotein in the ILN. This is meant to allow spread of the rabies virus an additional synaptic step to the DCN neurons that are presynaptic to the ILN cells that are presynaptic to the three striatal populations. The fundamental problem with this approach is that nonspecific expression of G in the ILN (or probably more broadly in the thalamus, because it would be difficult to restrict an injection to the ILN, and we can't really see clear GFP signal in the images) means that G is expressed not only in any thalamic neurons that are presynaptic to the starting cells in striatum, as intended, but also in any thalamic neurons that are presynaptic to those directly-presynaptic thalamic neurons, and so on. So then the virus would spread through those indirectly-connected thalamic cells and label *their* inputs, as well as the inputs to the actually monosynaptically connected ILN cells. This approach (which I know has been used by others) would only make sense if there were no connections between thalamic

neurons. In that case, the channels would indeed be separate. Similarly, the "trisynaptic" version (Figure 5) also requires that there are no connections between DCN neurons to be valid.

The reviewer raised three important points:

First, regarding the **multisynaptic nature** of the viral tracing when G-protein is provided not just in started but also relay cells.

Second, regarding the ability of evaluating **GFP marking** of cells that express cre-recombinase in our tracing experiments.

Third, regarding the fact to what extent GFP (and cre) expression relates to **expression of G protein**.

We appreciate the limitation of this "**multisynaptic**" **use of the rabies tracing** method and in the original manuscript we had been cautious to not call the connectivity assessed with this method "disynaptic" but rather referred to multisynaptic tracing – which is similar to more classical use of viruses that are not monosynaptically restricted.

First, we want to note that we complement the multi-synaptic tracings with several experiments that are not affected by the limitation raised by the reviewer:

- (a) in Figure 1 we combine anterograde marking of DCN axons and retrograde marking from the striatum. This experiment demonstrates that DCN-derived axons form varicosities containing vGluT2 on ILN neurons back-labeled from the striatum.
- (b) in Figure 2 we use anterograde trans-neuronal WGA-cre delivered from the DCN and demonstrate that this marks cells in ILN as well as axonal projections to the striatum (as well as specific cortical areas).
- (c) In the revised manuscript, we now added new data demonstrating that indeed stimulation of ChR2-expressing axons derived from the DCN elicits synaptic currents in ILN neurons. This demonstrates that (at least some of) the synaptic boutons seen in Figure 1 are indeed functional synapses (**new data in Figure S1c**).

Second, we appreciate that the multisynaptic experiments take up a significant amount of space in the manuscript (which was largely a function of the space needed to display the results). Knowing the weaknesses of this method, we now moved part of this data to the supplement (**Figure S3**) and more explicitly noted that the limitations of the interpretation in the text.

Third, the degree to how much our multi-synaptic tracing results are distorted by additional trans-synaptic viral transfers taking place between thalamic nuclei is a function of time and the connectivity between neurons in this area (which is currently unknown). To obtain an estimate of viral transfer between thalamic nuclei, we compared the viral marking of thalamic neurons in our multi-synaptic tracing experiments with previously published mono-synaptic tracing from D1R-cre, A2A-cre, and ChAT-cre neurons in the striatum ((Guo et al., 2015; Watabe-Uchida et al., 2012). Notably, we find the distribution of labeled thalamic neurons to be very similar to what was obtained in these previous monosynaptic studies. We now discussed this comparison in the text.

Also, the authors do not show the results of the much simpler monosynaptic tracing experiment that is the precursor of the "disynaptic" one, in order to see where in the ILN/etc. the different thalamic inputs to the three striatal populations are. Understood that getting at the disynaptic, rather than monosynaptic, inputs is the point of the paper, but showing the results of the simpler experiment would help in interpretation of the more complex ones.

We have performed the requested monosynaptic retrograde tracing from the ChAT interneurons in the striatum (as this had been reported only by one previous study and as the observation of cerebello-striatal connectivity to ChAT neurons is an important aspect of our study). In these experiments we replicated the previous results from Guo et al., 2015. We further provide a side-by-side comparison of the thalamic nuclei marked by our mono- and multisynaptic tracings (**new Figure S2**). Still, we cannot exclude that our multisynaptic tracings are affected at the level viral cross-over between neuronal sub-populations within these nuclei. We have clearly noted this in the text. Nevertheless, we note that viral tracing for mapping cerebello-striatal connections in mice have never been reported for mice. This connectivity has only been examined with multisynaptic viruses in primates which are more limited with respect to the interpretation. There, hypotheses about the mono- versus di- or trisynaptic connectivity was purely based on timing of viral transfer. We feel that (in combination with the experiments that are not affected by multisynaptic passage, see above) our experiments do add valuable information as the starter cells in the striatum are genetically defined. At the same time, we agree that it is important to carefully interpret the data noting the caveat of potential additional intra-thalamic trans-synaptic passages.

A possible approach at this point might be, having developed hypotheses about the locations in DCN of the neurons disynaptically connected to the striatal cell groups, to simply use monosynaptic tracing to label their direct inputs in ILN, in combination with injection of AAV-ChR2 in DCN, then do whole-cell recordings ex vivo with optogenetic activation of the (cut) DCN axons to see whether a given region of DCN indeed connects to the cells that are presynaptic to a given striatal cell type.

We have now done these recordings. We do find that ChR2-based stimulation of DCN axons indeed evokes synaptic responses in ILN neurons. This **new data was added to Figure S1c**.

Re detection of the GFP signal to see where G is really expressed, the best thing for tracking Cre expression would have been to do the experiments in Cre reporter mice (Ai14, etc., crossed to the Cre lines, of course), which would allow sensitive detection of where Cre was expressed at high enough levels to have a functional effect; second best would have been to immunostain for GFP) in order to see what cells are expressing G (EGFP-Cre being a proxy for G expression, given that immunostaining for G is evidently an unsolved problem in the field).

We apologize that the GFP-expression was invisible in the lower magnification images shown in the original version of the manuscript (it was obscured by the high level of mCherry fluorescence derived from the rabies vectors). We use a GFP marker in two configurations: First, as part of AAV2-DiO-G-TVA. GFP expression derived from this construct is (in our hands) indeed too dim to reliably detect in rabies co-infected cells. Second, we use AAV2-cre-GFP. In this case GFP expression is robustly detected. We now include individual channels and higher

magnification views of AAV2-cre-GFP infected cells to make GFP-fluorescence visible to the reader (**new Figure S2b and Figure S3c,d,e**).

We further raised antibodies against the rabies G-protein to test whether GFP-expression is indeed a reliable predictor of the cre-dependent G-expression. We quantified GFP/RV-G double-positive cells and conclude that 82.5% of GFP-positive cells in the ILN injections indeed express detectable G protein. We understand that it is difficult (impossible?) to predict how G protein detection relates to cells being competent of mediating efficient viral transfer. Regardless, we feel that this is one step in the direction of solving this problem and we hope that this new antibody will be a useful tool for the field (**see new data in Figure S4**). In fact, our colleagues in Basel already published data obtained with this reagent in a recent paper (Xu et al., 2016).

In a similar vein, "We quantified GFP-positive striatal neurons in ChAT-cre mice (N=3) to test if there 141 was any undesired retrograde transport of AAV2-hSyn-cre-GFP from ILN to ST. We did not 142 detect notable signal in these experiments" doesn't mean much, because we can't see the GFP anyway, and Cre would be effective at a very low level. This is a relatively minor point, because I would not expect a great deal of retrograde transport of AAV2.

See discussion in the point above: in this case we can indeed reliably detect GFP-expression. We agree that controlling the potential retrograde transport of AAV2 is a minor point but felt it was worth noting that we have attempted to probe whether such unexpected retrograde transport takes place.

Very minor comments:

We fixed the spelling mistakes noted in the review – thank you for pointing them out.

Reviewer #3 (Remarks to the Author):

The manuscript by Xiao and colleagues shows disynaptic anatomical connection between the cerebellum and the striatum, through the thalamus. They show that this connection reaches D1 and D2 MSNs, and also very strongly ChAT-positive interneurons. The authors show that if they silence DCN with DREADDs, activity-dependent gene expression decreases significantly in ChAT neurons in DMS. However, the authors do not show that this effect is mediated through the thalamus. The main issue of this study is in the last figure - behavior. The effects are weak, and the exclusion/division criteria applied of 50% is arbitrary. The authors should show an overall correlation (not only of animals with more than 50% infection, before applying any exclusion criteria). Also, the authors claim that the T-maze alternation task is striatal-dependent, but that is not clear. Finally, it would be important to show that the behavioral effects depend indeed on the disynaptic connection from DCN to striatum via thalamus.

So overall, the proof of disynaptic anatomical connection from DCN to striatum is strong, but the functional characterization of this disynaptic connection from a behavioral standpoint is weak and should be improved. As it stands, the behavioral

conclusions are not supported.

We thank the reviewer for the detailed evaluation of the work. We have now carefully evaluated options to strengthen the behavioral experiments, in particular to assess the specific requirement of the cerebellum-derived connections to the striatum. To address this, we conducted a series of new local silencing experiments in the striatum. Thus, we selectively manipulated the activity of striatal projections from ILN cells that are innervated from the DCN. We used an intersectional viral strategy to drive hM4Di in thalamic cells that are contacted by DCN axons (expressing WGA-cre, as previously established for the anatomical tracing experiments). We then cannulated mice in the dorsal striatum and locally infused CNO to selectively silence the striatum-projecting axons that relay DCN inputs to the ILN. This method was adopted from previous work by Sternson and colleagues who had applied this previously to hypothalamic outputs (Stachniak et al., 2014). We then asked whether the silencing of the ILN-derived axons would impair performance of the mice in the T-maze task. We found that this striatal silencing indeed recapitulated the behavioral phenotype observed for silencing of DCN neurons (**new Figure 7**). These new findings strongly support the hypothesis that DCN outputs regulate striatal control of alternation behavior through connectivity via thalamic nuclei.

Minor points:

- In the confocal images, the authors should show overlap using z-stack.
We have now added additional optical sections in Figure 1f,g to better visualize the apposition of synaptic markers on the retrogradely labeled ILN cells.
- Figure 6e/f panels seem to be reversed
Indeed. We corrected this and thank the reviewer for pointing out this error.
- The authors do not seem to normalize the cell counts for the number of starter cells
Given that some of the starter cell populations (in particular the MSNs) in the striatum are very dense it is difficult to accurately quantify starter cell numbers in these experiments. Thus, we mapped the relative distribution of retrogradely labeled cells for all animals. This method has been frequently used by others. We have now more clearly explained this in the figure legends.
- Methods are missing for quantification of cell counts for Fig 2, 3
We have now added this to the method section.
- Fig 2h, they should show D1 stainings as well, only showing D2 and ChAT (in supp?)
PENK and ChAT are good markers for the D2R+ and the ChAT neuron populations. Unfortunately, we have not found a suitable antibody that would enable us to probe the D1R population in our preparations. Thus, our conclusion that D1R+ cells are targeted by the cerebello-striatal connectivity is based only on the transsynaptic rabies tracing (whereas for the other populations we have been able to demonstrate this connectivity with two independent methods). We have now clarified this in the text.

Stats/uncertainties:

- Figures 3e, 4e, 7g, error bars not explained

We have now added this information in the figure legends.

Reviewer #4 (Remarks to the Author):

This manuscript explores the organization and function of the di-synaptic connection from the deep cerebellar nuclei (DCN) to the striatum in mice. Understanding how cerebellar outputs influence striatal activity is highly relevant to the field of neuroscience because of wide implications for motor and cognitive functions, as well as for multiple disorders. As discussed below, the authors claim to provide three independent lines of evidence for cerebellar influence on striatal activity, via the intralaminar nuclei (ILN) of the thalamus.

We thank the reviewer for the thorough review of the manuscript and the constructive suggestions.

1. The authors use anatomical tracing techniques and identify di-synaptic projections from the DCN to multiple classes of neurons in the striatum. The anatomical results are not entirely novel, as the authors imply. However, they do expand on the understanding of the di-synaptic projection from the DCN to the striatum.

First, previous findings in primates have shown that inputs to the striatum originate from all of the DCN (Hoshi et al., 2005). Thus, the fact that all the DCN send di-synaptic projections to the striatum is not novel and thus, the authors' focus on the medial and interposed nuclei is limiting. This limitation raises the question of how projections from these nuclei compare with projections from the lateral nucleus. This question is particularly relevant given that the lateral nucleus is the source of a majority of cerebellar output.

The pioneering studies by Strick and colleagues indeed provided evidence for connectivity from several DCN subnuclei but in rodents only connectivity from lateral nucleus had been studied in detail. The primate studies were conducted with polysynaptic viruses. This means, these tracing tools do not enable a reliable mono-synaptic restriction of the tracer, tracing can only be initiated from a brain region rather than a genetically-defined cell type, and can cross between intermediary cells within a nucleus in case they are connected. Conclusions about mono-, di-, tri-synaptic connectivity are based on latency of viral labeling. These challenges can be (at least partially) overcome in the genetically more accessible rodent model and we feel that our study adds to this important previous literature in two ways. First, mouse genetics provides unique tools to restrict tracing to specific cells based on their cell type identity. Thus, in our tracing experiments we could unambiguously initiate tracing from D1R-, D2R-, and ChAT neurons in the striatum. This cell type-specificity provides important insights into the targets of cerebello-striatal connectivity. Second, while basal ganglia, DCN and cerebellar structures are highly conserved from rodents to primates their sizes and sub-compartmentalization are significantly different between primates and rodents. Thus, it has been discussed whether contributions of the cerebellar outputs to cognitive function might be fundamentally different between rodents and primates (Galliano and De Zeeuw, 2014; Herculano-Houzel, 2010). Regardless, in response to the reviewers comment we have toned down the text regarding the novelty of this aspect of our study.

Second, studies in mice have shown that stimulation of the DCN modulate a variety of neuronal types in the striatum, including medium spiny neurons (MSNs), fast spiking interneurons, and potentially also cholinergic (ChAt) interneurons (Chen et al., 2014). Using cell-type specific tracing techniques, the authors show that all of the DCN send di-synaptic projections to MSNs that express both D1 and D2 receptors. This aspect of the results is new and interesting; it suggests that both the direct and indirect pathways through the striatum may be influenced by cerebellar activity in mice (see Bostan & Strick 2018 for a discussion of potential selectivity of cerebellar outputs to the indirect pathway).

The Chen et al study (Chen et al., 2014) indeed suggests that stimulation of DCN modifies several classes of neurons. However, they could not distinguish D1R and D2R MSNs and could not make any conclusions about ChAT interneurons at all (as stated in their discussion “parameters only allow one to unambiguously distinguish fast-spiking interneurons and MSNs”). We agree with the reviewer that the observation of cerebello-striatal coupling to genetically-defined cell populations in the striatum as an important element of our study.

The most interesting finding in this manuscript is arguably the direct evidence for DCN di-synaptic projections to ChAt interneurons in the striatum. As the authors appear to agree, this finding encourages further exploration. Arguably, focus on the organization and function of the di-synaptic projection from the DCN to ChAt interneurons would provide a more cohesive theme for the manuscript. Although the authors do pursue related questions, their results provide limited insights into the functions of this pathway (see points 2 and 3).

To further strengthen this part of the manuscript, we now performed additional monosynaptic tracing experiments from ChAT interneurons in the dorsal striatum (**new data in Figure S2**). Regarding the functional relevance of this pathway we have now performed a local silencing experiment (see below) that – we feel – greatly strengthens our conclusions about the involvement of cerebello-striatal connections in the striatum-dependent learning task.

Another interesting result is that individual DCN send differential projections to specific cell types in the striatum (Figure 4). As the di-synaptic projections from all the DCN reach all the studies striatal cell types, the widespread location of Purkinje cells that project tri-synaptically to D2 MSNs (Figure 5) is not particularly surprising. Although the authors do not explore this issue, Figure 4 suggests some level of organization of di-synaptic projections to specific cells types in the striatum within individual nuclei. It would be useful if the authors provided insights into this organization. Furthermore, it would be informative to determine whether multi-synaptic projections from Purkinje cells to specific cell types in the striatum are (also) topographically organized. Such organization might provide some insights into the potential diversity of functions of the cerebello-striatal connection. Thus, it is unclear why the authors chose to only study the multi-synaptic projection from Purkinje cells to the striatum in the A2A-cre mice.

We agree with the reviewer that a further, sub-nucleus-specific analysis of trisynaptic connections would be interesting. However, the currently available viral tools do not allow for the necessary restriction of tracing to trisynaptic (versus multisynaptic) connections. We agree that our data reveal a differential back-labeling of DCN neurons. However, given that these experiments are based on polysynaptic tracing

(where cross-over between cells within the thalamic nuclei or within the DCN can not be excluded) we felt that we need to be cautious with the interpretation of this result (this is also a point highlighted by reviewer 2). For the same reason, we have not repeated the multistage tracing experiments for all three striatal starter lines but only reported the results for the A2A-cre -defined cells. Since a substantial technology development is necessary to strictly limit the viral tracing to only reveal trisynaptic connections, we feel that these additional experiments require an entirely new study in the future. We have now more explicitly noted this issue in the manuscript.

2. The authors use selective silencing of deep cerebellar nuclei to assess their effects on striatal ChAT interneuron activity. Previous work has shown interesting effects on markers of ChAT interneuron activity and related behaviors after lesions of ILN (e.g., Bradfield et al. 2013). As expected from this previous work, the authors show differential effects of DCN neuron silencing on markers of ChAT interneuron activity in the dorsomedial striatum (DMS) vs. dorsolateral striatum (DLS). However, because of the unconstrained injection sites in the anatomical studies performed here, the reason for differential effects of DCN on DMS vs. DLS remains unclear. Determining how individual DCN target specific cell types (including ChAT interneurons) in functionally different striatal regions would go a long way in clarifying the different functions of this pathway.

As discussed in the manuscript, our hypothesis that DCN outputs may modify ChAT interneuron activity and in particular the pS6rp marker was based on the work by Bradfield et al conducted in rats. However, there had been no evidence that suppression of endogenous DCN activity has a significant impact on the activity of striatal ChAT interneurons (in fact, even the impact on MSNs reported by Chen et al used exogenous stimulation of DCN outputs, and thus, represents a gain-of-function experiment). Thus, we feel that it is actually an important, novel finding. At present, we do not know the reason for the selective effect on DMS versus DLS ChAT interneurons. However, we hypothesize that this is a consequence of the different levels of pS6rp detected in these populations of ChAT cells “at baseline”: Without DCN manipulation, we find that pS6rp levels in DMS ChAT interneurons are higher than in DLS ChAT interneurons. This might be the reason why we do not observe a further reduction in DLS ChAT cells upon DCN silencing. We now discuss this possibility in the revised manuscript.

3. The authors investigate the behavioral effects of selective silencing of deep cerebellar nuclei on behavior. This is the least well-developed aspect of the paper and not particularly informative. First, the authors do not provide electrophysiological that the DCN have actually been silenced. Second, there is no behavioral evidence for functional silencing of the DCN. It is unclear why the mice were not evaluated on a task that is dependent on DCN function. Finally, the authors test the effects of silencing the DCN on “striatum-dependent” tasks and note that performance is reminiscent of mice with abnormal ChAT interneuron function. The authors’ interpretation of this data is a stretch for two main reasons: 1) the authors do not use tasks that have been shown to be ChAT interneuron dependent (e.g., Bradfield et al. 2013) in order to properly compare their results with previous findings, and 2) although the “forced alternation” T-maze task involves mice obtaining a reward, there are multiple possible routes for DCN outputs to influence behavior on this task that do not involve DCN projections to the striatum (e.g., projections to

cortex).

The behavioral task was modeled based on previous studies in mice that directly inactivated/ablated striatal ChAT interneurons and that reported activity of striatal ChAT interneurons engaged in a T-maze task (Atallah et al., 2014; Kitabatake et al., 2003). These are somewhat different from the task used by Bradfield and colleagues who performed their experiments in rats. Regardless, we agree with the reviewer that there are other output routes from the DCN that might influence the behavior.

To address this major point, we selectively manipulated the activity of striatal axons from ILN cells that are innervated from the DCN. To this end, we used an intersectional viral strategy to drive hM4Di in thalamic cells that are contacted by DCN axons (expressing WGA-cre as originally used in Figure 2 for anatomical tracing). We then locally infused CNO into the dorsal striatum and asked whether the silencing of the ILN-derived axons would impair performance of the mice in the T-maze task. We found that this indeed recapitulated the behavioral phenotype observed for silencing of DCN neurons (**new Figure 7**). These new findings strongly support the hypothesis that DCN outputs regulate striatal control of alternation behavior through connectivity via thalamic nuclei.

Literature cited:

- Atallah, H.E., McCool, A.D., Howe, M.W., and Graybiel, A.M. (2014). Neurons in the ventral striatum exhibit cell-type-specific representations of outcome during learning. *Neuron* 82, 1145-1156.
- Chen, C.H., Fremont, R., Arteaga-Bracho, E.E., and Khodakhah, K. (2014). Short latency cerebellar modulation of the basal ganglia. *Nat Neurosci* 17, 1767-1775.
- Galliano, E., and De Zeeuw, C.I. (2014). Questioning the cerebellar doctrine. *Prog Brain Res* 210, 59-77.
- Guo, Q., Wang, D., He, X., Feng, Q., Lin, R., Xu, F., Fu, L., and Luo, M. (2015). Whole-brain mapping of inputs to projection neurons and cholinergic interneurons in the dorsal striatum. *PLoS One* 10, e0123381.
- Herculano-Houzel, S. (2010). Coordinated scaling of cortical and cerebellar numbers of neurons. *Front Neuroanat* 4, 12.
- Kitabatake, Y., Hikida, T., Watanabe, D., Pastan, I., and Nakanishi, S. (2003). Impairment of reward-related learning by cholinergic cell ablation in the striatum. *Proc Natl Acad Sci U S A* 100, 7965-7970.
- Stachniak, T.J., Ghosh, A., and Sternson, S.M. (2014). Chemogenetic synaptic silencing of neural circuits localizes a hypothalamusmidbrain pathway for feeding behavior. *Neuron* 82, 797-808.
- Watabe-Uchida, M., Zhu, L., Ogawa, S.K., Vamanrao, A., and Uchida, N. (2012). Whole-brain mapping of direct inputs to midbrain dopamine neurons. *Neuron* 74, 858-873.
- Xu, C., Krabbe, S., Grundemann, J., Botta, P., Fadok, J.P., Osakada, F., Saur, D., Grewe, B.F., Schnitzer, M.J., Callaway, E.M., et al. (2016). Distinct Hippocampal Pathways Mediate Dissociable Roles of Context in Memory Retrieval. *Cell* 167, 961-972 e916.

Reviewers' comments:

Reviewer #1 (Remarks to the Author):

The authors have sufficiently addressed all my major concerns. Good work!

Reviewer #2 (Remarks to the Author):

The authors have not addressed my primary concern, which is that the "disynaptic" (not to mention the "trisynaptic") tracing they perform using rabies virus is basically uninterpretable.

As a minor point, the authors claim in their rebuttal that "in the original manuscript we had been cautious to not call the connectivity assessed with this method "disynaptic" but rather referred to multisynaptic tracing", but the title, abstract, section headers, and throughout the manuscript the term "disynaptic" is used. While there is other direct evidence for a disynaptic connection to striatum generally (e.g., the combined bulk-retrograde and anterograde labeling in figure 1), the tracing with rabies virus is central to the manuscript, leading readers to believe that it provides much of the evidence for, as stated in the abstract, "a broad disynaptic connectivity matrix from the deep cerebellar nuclei (DCN) to the dorsal striatum in mice. Cerebello-striatal connections arise from all deep cerebellar sub-nuclei and are relayed through intralaminar thalamic nuclei (ILN). In the dorsal striatum, these connections target medium spiny neurons, but also ChAT-positive interneurons", whereas it doesn't in its current form.

To remedy the current deficiencies, I had suggested a fairly straightforward set of experiments that are evidently within the capabilities of the authors and that would have provided direct evidence for the claim of disynaptic connections from DCN to each of the three populations in striatum (D1R+, A2a+, ChAT+ cells), namely to perform simple monosynaptic tracing from each of the three populations to fluorescently label the ILN neurons presynaptic to each, in mice also injected with an AAV-ChR2 in DCN, then confirm the existence of direct synapses from DCN to the labeled ILN neurons using ex vivo whole-cell recording and photostimulation of the cut axons. The authors instead simply did such ex vivo optogenetic confirmation of connectivity of DCN axons onto unlabeled ILN cells, which says nothing about the existence of a disynaptic connection from DCN to the three different striatal cell types.

It seems from the other reviewers' comments (e.g., "The most interesting finding in this manuscript is arguably the direct evidence for DCN di-synaptic projections to ChAT interneurons in the striatum.") that the claims are central to the novelty and significance of the manuscript and therefore need to be supported as suggested above or else removed along with almost all of the rabies viral tracing results, which are simply misleading.

Reviewer #3 (Remarks to the Author):

The authors did a good job addressing most of the concerns.

However, figures 6 I and J remain unchanged. We had expressed concern that a 50% infection rate is an arbitrary criteria, and it is not justifiable to do that. What if one considers 45%? Or 40%? Is 50% the only number that works?

Also, the new Figure 7 is important, but does the new figure 7i also use the same criteria as 6 i?

Presumably it is difficult to have more than 50% criteria here. The panels are arranged in the same way so the criteria should be the same (and not what works in terms of statistical significance).

Reviewer #4 (Remarks to the Author):

The manuscript by Xiao et al. provides an in-depth exploration of the organization of the di-synaptic connection from the deep cerebellar nuclei to striatal cells in mice. The authors performed additional experiments and analyses that satisfactorily addressed several important issues raised by the reviewers in response to the original version of the manuscript. Overall, the revisions to the manuscript help strengthen the author's claims regarding the relevance and function of cerebellar influence on striatal activity, via the intralaminar nuclei (ILN) of the thalamus.

Point-by-point response to reviewers' comments:

Reviewer #1 (Remarks to the Author):

The authors have sufficiently addressed all my major concerns. Good work!

We thank the reviewer for the appreciation of the work and the constructive suggestions made during the review process.

Reviewer #2 (Remarks to the Author):

To further address the concern of the reviewer that the rabies virus tracing experiments may be misinterpreted we have eliminated all statements suggesting or implying di- or trisynaptic connectivity when relating to our own experiments. We have further modified the title of the manuscript to implement the reviewers' requests.

Reviewer #3 (Remarks to the Author):

The authors did a good job addressing most of the concerns.

However, figures 6 I and J remain unchanged. We had expressed concern that a 50% infection rate is an arbitrary criteria, and it is not justifiable to do that. What if one considers 45%? Or 40%? Is 50% the only number that works?

We note that it is a standard step in such experiments to apply cut-offs to account for varying efficiencies in viral infection between animals. We further appreciate that the exact choice of the cut-off value is always arbitrary. We note that we made data from all animals available in Figure 6J. This figure demonstrates (independently of any cut-off) that there is a statistically significant correlation between the number of silenced DCN cells and the performance of the animal in the behavioral task. Thus, 50% is not "the only number that works".

Also, the new Figure 7 is important, but does the new figure 7i also use the same criteria as 6 i? Presumably it is difficult to have more than 50% criteria here. The panels are arranged in the same way so the criteria should be the same (and not what works in terms of statistical significance). The experiment is different in that the limiting factor is the intersectional turning on of hM4Di in the ILN rather than the infection of DCN neurons. We did not use any cut-off for DCN infection in this experiment as (building on the knowledge gain in the previous experiments) we used higher titer AAV preparations that yielded overall high efficiency infection of DCN neurons in all animals. We note that in the absence of a cut-off for minimal infection there is no concern regarding introduction of bias for "what works for statistical significance". The only exclusion criterion applied in this set of experiments was the positioning of the cannulae in the striatum. This is essential since mis-placed cannulae will result in CNO reaching other brain areas than the striatum and would not allow for specific conclusions about cerebello- striatal connectivity. We now more clearly explained this in the figure legend and the methods.

Reviewer #4 (Remarks to the Author):

The manuscript by Xiao et al. provides an in-depth exploration of the organization of the di-synaptic connection from the deep cerebellar nuclei to striatal cells in mice. The authors performed additional experiments and analyses that satisfactorily addressed several important issues raised by the reviewers in response to the original version of the manuscript. Overall, the revisions to the manuscript help strengthen the author's claims regarding the relevance and function of cerebellar influence on striatal activity, via the intralaminar nuclei (ILN) of the thalamus.

We thank the reviewer for the thoughtful feedback on the original version of the manuscript as well as the appreciation of the experiments added in the revision.